# *Candida albicans* commensalism in the oral mucosa is favoured by limited virulence and metabolic adaptation

Christina Lemberg[1,2☯], Kontxi Martinez de San Vicente[1,2☯], Ricardo Fróis-Martins[1,2☯], Simon Altmeier[1,2], Van Du T. Tran[3], Sarah Mertens[1,2], Sara Amorim-Vaz[4], Laxmi Shanker Rai[5], Christophe d'Enfert[5], Marco Pagni[3], Dominique Sanglard[4☯], Salomé LeibundGut-Landmann[1,2☯]*

1 Section of Immunology, Vetsuisse Faculty, University of Zürich, Zürich, Switzerland, 2 Institute of Experimental Immunology, University of Zürich, Zürich, Switzerland, 3 Vital-IT Group, SIB Swiss Institute of Bioinformatics, Lausanne, Switzerland, 4 Institute of Microbiology, University of Lausanne and University Hospital Center, Lausanne, Switzerland, 5 Institut Pasteur, Université de Paris, INRAE, USC2019, Unité Biologie et Pathogénicité Fongiques, Paris, France

☯ These authors contributed equally to this work.
* salome.leibundgut-landmann@uzh.ch

**Data Availability Statement:** Transcriptomic data are deposited on the NCBI BioProject PRJNA491801 (https://www.ncbi.nlm.nih.gov/

## Abstract

As part of the human microbiota, the fungus *Candida albicans* colonizes the oral cavity and other mucosal surfaces of the human body. Commensalism is tightly controlled by complex interactions of the fungus and the host to preclude fungal elimination but also fungal overgrowth and invasion, which can result in disease. As such, defects in antifungal T cell immunity render individuals susceptible to oral thrush due to interrupted immunosurveillance of the oral mucosa. The factors that promote commensalism and ensure persistence of *C. albicans* in a fully immunocompetent host remain less clear. Using an experimental model of *C. albicans* oral colonization in mice we explored fungal determinants of commensalism in the oral cavity. Transcript profiling of the oral isolate 101 in the murine tongue tissue revealed a characteristic metabolic profile tailored to the nutrient poor conditions in the stratum corneum of the epithelium where the fungus resides. Metabolic adaptation of isolate 101 was also reflected in enhanced nutrient acquisition when grown on oral mucosa substrates. Persistent colonization of the oral mucosa by *C. albicans* also correlated inversely with the capacity of the fungus to induce epithelial cell damage and to elicit an inflammatory response. Here we show that these immune evasive properties of isolate 101 are explained by a strong attenuation of a number of virulence genes, including those linked to filamentation. De-repression of the hyphal program by deletion or conditional repression of *NRG1* abolished the commensal behaviour of isolate 101, thereby establishing a central role of this factor in the commensal lifestyle of *C. albicans* in the oral niche of the host.

bioproject/?term=PRJNA491801). All other data linked to this manuscript are publicly available on Zenodo (https://doi.org/10.5281/zenodo.6393169).

**Funding:** CL was funded by a grant from the Swiss National Science Foundation to SLL (grant N° 310030_166206, www.snf.ch). KMSV was funded by a grant from the Swiss National Science Foundation to SLL, SD and CdE (grant N° CRSII5_173863, www.snf.ch). SA and SAV were funded by a grant from the Swiss National Science Foundation to DS, MP and SLL (grants N° 310030_166206, CRSII5_173863 and CRSII3_141848, respectively, www.snf.ch). RFM was funded through the European Union's Horizon 2020 research and innovation programme under the Marie Sklodowska-Curie action, Innovative Training Network (FunHoMic; grant N° 812969, www.funhomic.eu), to CdE and SLL. LSR's post-doctoral fellowship was funded through a Fondation pour la Recherche Médicale grant to CdE (FRM, DBF20160635719, www.frm.org/). SIB partly funded by the Swiss Federal Government through the Federal Office of Education and Science (https://www.bfs.admin.ch/bfs/en/home/statistics/education-science.html). Work in CdE's laboratory is supported by a grant from the Agence Nationale de la Recherche (ANR-10-LABX-62-IBEID). The funders had no role in study design, data collection and analysis, decision to publish, or preparation of the manuscript.

**Competing interests:** We have read the journal's policy and the authors of this manuscript have the following competing interests:Simon Altmeier is now an employee of Novartis. His findings presented in this paper are solely his views and are independent from Novartis work. The other authors have no competing interest to declare.

## Author summary

The oral microbiota represents an important part of the human microbiota and includes several hundreds to several thousands of bacterial and fungal species. One of the most prominent fungus colonizing the oral cavity is the yeast *Candida albicans*. While the presence of *C. albicans* usually remains unnoticed, the fungus can under certain circumstances cause lesions on the lining of the mouth referred to as oral thrush or contribute to other common oral diseases such as caries. Maintaining *C. albicans* commensalism in the oral mucosa is therefore of utmost importance for oral health and overall wellbeing. While overt fungal growth and disease is limited by immunosurveillance mechanisms during homeostasis, *C. albicans* strives to survive and evades elimination from the host. Here, we show that while commensalism in the oral cavity is characterized by a restricted fungal virulence and hyphal program, enforcing filamentation in a commensal isolate is sufficient for driving pathogenicity and fungus-induced inflammation in the oral mucosa thwarting persistent colonization. Our results further support a critical role for specialized nutrient acquisition allowing the fungus to thrive in the nutrient poor environment of the squamous epithelium. Together, this work revealed key determinants of *C. albicans* commensalism in the oral niche.

## Introduction

*Candida albicans* is a prominent member of the human mycobiota in the oral, gastrointestinal and vaginal mucosa. As a pathobiont, *C. albicans* can also cause infections reaching from mild superficial manifestations to more severe and frequently life-threatening systemic candidiasis [1]. Infections occur primarily in immunocompromised individuals, with mucocutaneous candidiasis developing most frequently in patients with acquired or inherited defects in T cell and IL-17 immunity [2]. While intact host defences are critical for preventing *C. albicans* overgrowth and tissue invasion, the balance between commensalism and disease also depends on the virulence state of the colonizing fungal isolate. The species of *C. albicans* displays a large genetic diversity which translates in phenotypic differences between isolates. Although the genetic landscape of *C. albicans* has been described in some details [3], the functional consequences of these intraspecies variations at the host interface are less clear. Experimental infection studies in previously *C. albicans*-naïve mice with small sets of fungal isolates revealed diverse outcomes in terms of host survival (in the model of systemic candidiasis) or in terms of persistence of colonization (in the model of oropharyngeal candidiasis) [4–7]. The molecular basis for the commensal versus pathogenic lifestyle of genetically and phenotypically distinct natural isolates of *C. albicans* remains ill defined. Some pathogenicity traits of *C. albicans* have been identified in the gastrointestinal tract [8]. The generation of isogenic mutants of the highly virulent reference isolate SC5314 and within-host evolution experiments with SC5314 allowed the identification of genetic determinants that govern the ability of *C. albicans* to inhabit the gastrointestinal tract [9–13]. While non-filamentous morphologies are generally good colonizers, hyphae are less fit in the gastrointestinal tract but instead may be better suited for epithelial translocation and invasive infections [10,14,15]. Beyond morphology, fungal adhesion and host cell damage induction can also affect gut colonization efficiency [16]. Much less is known about the fungal determinants that enable *C. albicans* colonization of the oral mucosa, an important niche of the fungus [17] and a frequent site of *C. albicans*-mediated disease manifestation [18].

The oral cavity differs significantly from the gastrointestinal tract and other commensal niches of *C. albicans* with respect to physico-chemical properties such as pH and $O_2$ content, the nature of the lining epithelium and the composition of the microbiota [19]. During oropharyngeal candidiasis (OPC) in mice, which are not naturally colonized with *C. albicans*, virulent isolates of *C. albicans* trigger a strong inflammatory host response that results in rapid clearance of the fungus from the oral mucosa. In contrast, low virulent isolates elicit only a limited host response which coincides with fungal persistence in the stratum corneum [7]. In the latter case, commensal colonization of fully immunocompetent and non-antibiotically treated wild type mice can last for over a year [20].

While the differential response of the host to commensal versus pathogenic isolates of *C. albicans* was dissected in some detail, the fungal factors that determine the commensal lifestyle of *C. albicans* in the oral niche remain to be explored. Using the prototypical commensal isolate 101 characterized by long persistence in the OPC mouse model, the objective of this study was to determine how the fungus survives and evades the host in this specific tissue compartment. Transcript profiling of the isolate inside the host tissue revealed hallmarks of commensalism in the oral niche. High expression levels of the transcription factor *NRG1* was identified as a determining factor in the commensalism of isolate 101, the relevance of which we confirmed by complementary genetic approaches.

## Results

### *C. albicans* isolate 101 displays reduced filamentation compared to SC5314

The commensal *C. albicans* isolate 101 efficiently and persistently colonizes the oral epithelium of wild type C57BL/6 mice when administered sublingually. Fungal elements locate to the stratum corneum where they persist without penetrating deeper epithelial layers of the stratified epithelium [7]. This is in contrast to the highly virulent isolate SC5314, which causes damage in the oral epithelium and invades the stratified epithelium to deeper layers. To determine the characteristics of fungal commensalism in the oral niche, we first assessed the morphology of isolate 101 on PAS-stained tissue sections and after isolation of the fungus from the tongue by KOH (**Fig 1A and 1B**). The fungus was predominantly in the hyphal form, whereby hyphae appeared to be slightly shorter in comparison to those of the isolate SC5314. By histology we also observed some pseudohyphae.

Because quantification of filament length is hardly possible *in vivo* on tissue sections, we examined filamentation *in vitro* in serum containing medium. Filaments of isolate 101 were consistently shorter than those of SC5314, in both presence or absence of oral keratinocytes (**Fig 1C–1H**). The morphogenetic defect of isolate 101 was most pronounced when cultured on Spider agar (**Fig 1I**). The commensal phenotype of isolate 101 was thus associated an impairment in filamentation, although the isolate was able to form hyphae, especially at the interface with the host.

### Isolate 101 is less invasive than SC5314

In line with the impaired filamentation phenotype, isolate 101 was also less invasive in YNB agar containing BSA when compared to SC5314 (**Fig 2A**). Similarly, when probing invasion of TR146 oral keratinocytes, a reduced proportion of 101 hyphae invaded host cells (**Fig 2B**).

Assaying fungal invasion of cultured keratinocyte monolayers monitors the invasion of *C. albicans* filaments into individual cells, while colonization of the stratified epithelium of the oral mucosa comprises both intra- and intercellular invasion processes. Probing the invasive capacities of isolate 101 in a fully differentiated three-dimensional skin model reproduced the situation in the oral epithelium *in vivo*, whereby isolate 101 remained mostly restricted to the

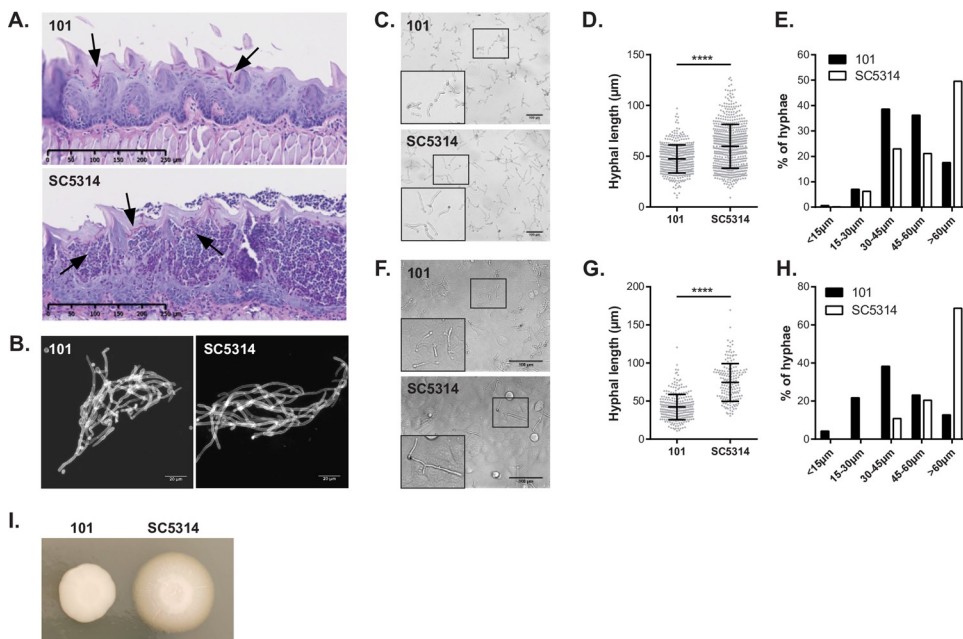

**Fig 1. C. albicans isolate 101 displays reduced filamentation compared to SC5314. A.-B.** C57BL/6 WT mice were infected sublingually with isolate 101 or SC5314. Histology images in **A** depict PAS-stained tongue sections on day 1 post-infection. Scale bar = 250 μm. Arrows point to fungal elements. Fungal elements isolated at the same time point by KOH were visualized with calcofluor white in **B**. Scale bar = 20 μm. Data are representative of at least 2 independent experiments. **C.-H.** *C. albicans* isolates 101 and SC5314 were grown for 3–4 hour in F12 medium (**C-E**) or on top of a monolayer of TR146 cells in F12 medium (**F-H**). Representative microscopy images are shown in **C** and **E**. Scale bar = 100 μm. Framed areas are magnified in the bottom left corner. Panels **D** and **G** show quantification of filament length. Each dot represents an independent hyphal element; the mean ± SD is indicated. In **E** and **H**, fungal elements were grouped according to their length and the proportion of fungal elements in each group is shown. Data are representative of at least 3 independent experiments for each condition. **I.** Representative image of isolates 101 and SC5314 grown on Spider agar for 5 days at 30˚C.

outermost keratinized layer, while SC5314 penetrated deep into the tissue, reaching even the supporting fibroblasts that underlie the epithelium in this model (**Fig 2C**). This finding indicated that the localization of *C. albicans* inside the stratified epithelium is an intrinsic property of the isolate in interaction with epithelial cells. This localization was independent of immune cells that represent an integral component of the mucosal tissue *in vivo*. The less invasive phenotype of isolate 101 in reconstituted human epidermis coincided with its low capacity to induce epithelial cell damage, which was assessed by LDH release assay (**Fig 2D**), and production of inflammatory cytokines (**Fig 2E and 2F**), similarly to the situation in keratinocyte monolayer cultures (**Fig 2G and 2H**).

## Altered interaction of isolate 101 with keratinocytes does not curb the initial infection process in the oral epithelium

To assess whether the observed phenotypic differences between isolates 101 and SC5314 affected their initial colonization of the host tissue, we tested their interaction with the oral epithelium in a competitive infection setting. To distinguish the two isolates after co-infection, we used a mCherry-expressing version of isolate 101 [21] and a GFP-expressing version of SC5314 [22], mixed in equal amounts (**S1A Fig**). After 1 day, both isolates were recovered from the infected tongue at a 1:1 ratio, comparable to the ratio in the infection inoculum, indicating that both isolates seeded the oral epithelium with comparable efficiency (**Fig 3A**),

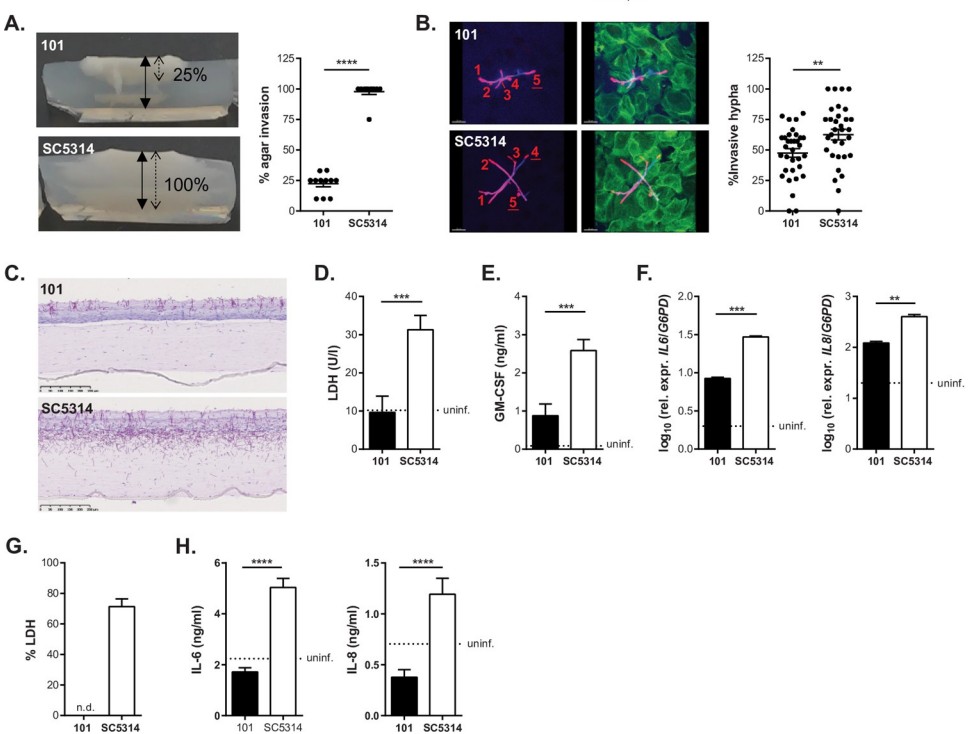

**Fig 2. Isolate 101 is less invasive than SC5314. A.** Representative images (left) and quantification (right) of the invasion depth of isolates 101 and SC5314 in YNB agar supplemented with BSA after 7 days of incubation at 37°C and 5% $CO_2$%. Agar invasion was calculated as the % of invasion depth (dotted arrow) relative to the overall thickness of the agar (solid arrow). Data were pooled from 4 independent experiments with 2–3 samples each. The mean±SEM is indicated. Similar results were obtained with YNB agar supplemented with glucose or casamino acids. **B.** Representative images (left) and quantification (right) of TR146 cell monolayers incubated with isolate 101 or SC5314 for 4 hours at 37°C and 5% $CO_2$. Extracellular fungal parts were stained with an anti-*Candida* antibody (red), full fungal elements were visualized by calcofluor white staining (blue), and keratinocytes were stained with phalloidin (green). The counted number of total hyphae (red numbers) and of invasive hyphae (underlined red numbers) are indicated on the representative images. Each symbol in the quantification is the % invasion per field analysed with an average of 8 hyphae per field (a total of 32 fields were analysed per isolate with 240 hyphae for SC5314 and 283 hyphae for 101). The mean±SEM is indicated for each group. Data were pooled from 2 independent experiments. **C.-E.** Reconstituted human epidermis (RHE) was infected with isolate 101 or SC5314 (bottom) for 24 hours. **C** shows PAS-stained histology sections. Scale bar = 250 μm. Cellular damage was assessed by LDH release assay in **D**, GM-CSF concentrations were determined in the supernatant by ELISA in **E**, and IL-6 and IL-8 transcript levels were quantified by RT-qPCR in **F**. Bars are the mean+SD of 4 (**D**, **E**) or 2 (**F**) replicate wells from a single experiment. **G.-H.** Monolayers of TR146 keratinocyte were infected with isolate 101 or SC5314 for 24 hours. Epithelial cell damage was assessed by LDH release assay in **G**, and IL-6 and IL-8 cytokine levels were quantified in the supernatant by ELISA in **H**. Bars are the mean+SD of 4 replicate wells. Data are representative of at least 2 independent experiments.

confirming previous findings from single-strain infections [7]. The initial steps of the infection process seem thus not to be affected by differences in the isolates' capacity to filament or to interact with keratinocytes *in vitro* (invasion and damage induction).

In single infections, the cellular damage caused by high-virulent isolates such as SC5314 triggers an acute inflammatory response in the infected epithelium characterized by rapid production and release of pro-inflammatory cytokines by epithelial cells, including IL-1α and chemokines, which trigger the recruitment of neutrophils and inflammatory monocytes from the circulation to the site of infection [23,24]. A consequence of the acute host response in the oral mucosa is the rapid elimination of the fungus from the experimentally infected host. In contrast, low-damaging isolates which trigger a limited or no inflammatory response upon colonization/infection persist in the epithelium over time [7,20].

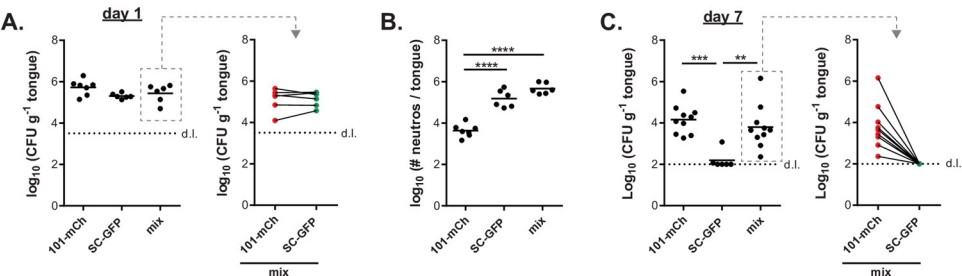

**Fig 3. Isolate 101 persists in the oral mucosa even in presence of SC5314-induced inflammation. A-C.** C57BL/6 WT mice were infected sublingually with a 1:1 mixture of isolates 101-mCherry (101-mCh) and SC5314-GFP (SC-GFP) or with either isolate alone. On day 1 (**A**) and on day 7 (**C**) the total tongue fungal burden was quantified (left panels). From the co-infected animals, the colonies were further analysed for mCherry- and GFP-expression (right panels). Quantification of tongue-infiltrating neutrophils by flow cytometry on day 1 post-infection is shown in **B**. Each symbol represents one animal. The mean of each group is indicated. Data in A and B are pooled from 2, data in C are pooled from 3 independent experiments. d.l., detection limit. Please see also **S1 Fig**.

Therefore, the question arose whether in a co-infection setting, where isolate 101 is exposed to an inflammatory environment induced by SC5314, it would be cleared from the oral mucosa, similarly to SC5314 itself. We confirmed that the overall inflammatory response was indeed comparable in the co-infection setting as in a single infection with SC5314 (**Figs 3B and S1B**). We then assessed the fungal burden in the tongue by day 7 post-infection, when SC5314 is cleared in a single infection setting [25,26]. Tongue fungal counts in co-infected animals remained high and were comparable as in isolate 101 single-infected mice (**Fig 3C**). Enumeration of the colonies based on their fluorescent label revealed that all were expressing mCherry and hence identified as 101, while no SC5314 colonies were recovered (**Fig 3C**). Together, this indicates that, even in a co-infection setting, isolate 101 persists in the oral mucosa and resists the inflammatory environment.

## Persistence of isolate 101 in the oral mucosa is associated with a reduced virulence profile

To obtain a more comprehensive and unbiased picture of the properties of isolate 101, we acquired the transcriptomes of isolates 101 and SC5314 as a reference within the oral mucosa at three consecutive time points (day 1, day 3, day 7 post-infection). However, due to its rapid clearance from the oral mucosa the analysis of isolate SC5314 transcriptome was limited to day 1 post-infection. We also included samples from *in vitro*-induced hyphae of both isolates, whereby we spiked this RNA into RNA isolated from naïve tongues.

Fungal RNA was enriched using a hybridization technology aimed at enriching fungal RNAs within infected host tissues where they usually account for only a small fraction of the overall RNA [27]. We verified that the baits used for RNA enrichment, which were designed on the basis of strain SC5314, or usage of SC5314 genome assembly for aligning sequencing data did not introduce a bias (see details in the methods section, **S2A–S2D Fig and S1 Table**). Several hundreds of genes were globally up- and downregulated under the chosen conditions (**S2E Fig and S2 Table**). Principal component analysis showed distinct transcript profiles of isolates 101 and SC5314, both under *in vitro* or *in vivo* conditions (**Fig 4A**). Moreover, the transcriptome of isolate 101 changed over time and became increasingly distinct from the one of SC5314 on day 1 post-infection (**Fig 4A**).

Qualitative differences of the transcriptional response of isolate 101 compared to SC5314 were reflected in a diminished expression of genes associated with filamentous growth, virulence, and immune activation, with differences increasing over time (**Figs 4B and S3**). Among

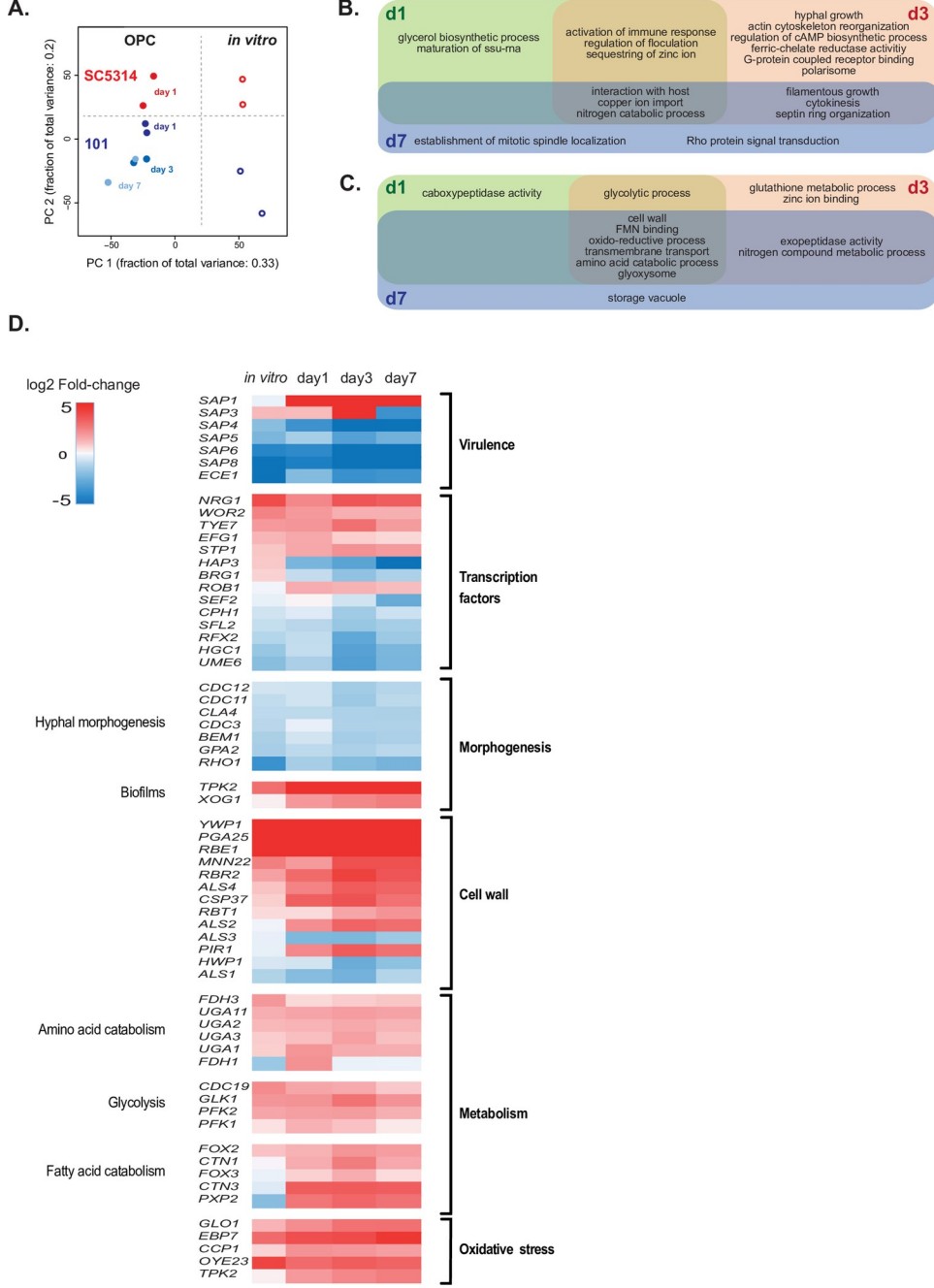

**Fig 4. Persistence of isolate 101 in the oral mucosa is associated with a reduced virulence profile and an altered metabolic signature. A.** PCA plot generated from the 5589 *C. albicans* genes that were detected in the RNAseq analysis. **B-C.** Venn diagram showing the GO terms associated with genes that were down-regulated (**B**) or up-regulated (**C**) in isolate 101 on day 1 (green), day 3 (red) and/or day 7 (blue) post-infection in comparison to SC5314 on day 1 post-infection. Only genes with p-values below 0.05 and log2 fold change $\geq 1$ and $\leq -1$ were considered. **D.** Heat map of selected differentially regulated genes linked to virulence, morphogenesis, cell wall and metabolism (p < 0.05). Genes are displayed by decreasing log2 fold-change for the '*in vitro*' condition (101 in vitro vs. SC5314 in vitro). day 1, 101 in vivo day 1 vs. SC5314 in vivo day 1; day 3, 101 in vivo day 3 vs. SC5314 in vivo day 1; day 7, 101 in vivo day 7 vs. SC5314 in vivo day 1. Please see also **S2** and **S3 Figs**.

the most strongly downregulated genes in isolate 101 featured genes encoding adhesins and invasins, namely *ALS1* and *ALS3* [28,29], aspartyl proteases, namely *SAP4*, *SAP6* and *SAP8* linked to deep tissue invasion [30], and the damage-inducing factor *ECE1* [24] (**Fig 4D**). In contrast, the increased expression of *SAP1* and *SAP3* by isolate 101 in the oral mucosa was consistent with the preferential residence of this isolate in the upper layers of the stratified epithelium [30]. Genes encoding transcription factors associated with filamentous growth including *CPH1*, *HGC1*, *BRG1* and *UME6* were also downregulated in isolate 101 as compared to SC5314. Interestingly, *NRG1* that encodes a repressor of the filamentation program [31] was strongly upregulated (**Fig 4D**). The difference in morphogenesis between the two isolates was further associated by differential regulation of cell wall and biofilm associated genes (**Fig 4D**).

Among the genes displaying an increased expression in isolate 101 relative to SC5314 were stress-response genes, especially those implicated in ROS detoxification and the response to oxidative stress (**Fig 4D**). This observation is in support of the persistent phenotype of the strain in the mucosal tissue, even under inflammatory conditions as during co-infection with an inflammation-inducing isolate (**Fig 3**).

Differential expression of virulence and morphogenesis genes between isolates 101 and SC5314 in the murine oral mucosa was largely conserved under *in vitro* culture conditions with serum (**Fig 4D**). RT-qPCR assays with RNA isolated from *C. albicans* after exposure to TR146 keratinocytes further confirmed the RNAseq data (**S4A and S4B Fig**). Overall, the transcriptomic data are well in agreement with the phenotypic data indicating that isolate 101 is less filamenting, less invasive, less tissue damaging and less immune activating than SC5314 (**Figs 1 and 2**) [7,20]. Of note, the changes in virulence gene expression increased over time and largely accounted for the time-dependent shifts in the transcript profile of isolate 101 (**Fig 4A**).

## Isolate 101 is metabolically adapted to the oral niche

In addition to the filamentation-related gene expression differences described above, we identified an unanticipated, albeit prominent metabolic signature in isolate 101. A large group of genes associated with metabolic processes, including amino acid and phospholipid catabolism, and with glycolysis and pyruvate metabolism were overexpressed in isolate 101 in comparison to SC5314 (**Fig 4C and 4D**). Surprisingly, these differences were apparent at every time point analyzed and both under *in vivo* and *in vitro* conditions (**Figs 4C, 4D and S4C**). Moreover, genes encoding membrane transporters, including oligopeptide transporters and glucose, dicarboxylate and monocarboxylate transporters were upregulated in the murine tissue at at least one time point analysed (**Fig 4C**). Taken together, these data suggested that isolate 101 is metabolically adapted and employs an efficient strategy for nutrient acquisition in the oral epithelium of the host where nutrients are limited.

In support of this notion, isolate 101 featured enhanced phospholipase activity in comparison to SC5314 when examined experimentally on original Prices' egg yolk agar (**S4D Fig**). Moreover, we observed that isolate 101 displayed an enhanced growth under conditions mimicking those in the oral mucosa, including the replacement of glucose as the carbon source by N-acetlyglucosamine (GlcNAc), an amide derivative sugar abundant in the saliva and in the cell wall and biofilm extracellular matrix of oral bacteria [32,33]. The enhanced growth phenotype was also visible on casamino acids, a mixture of free amino acids and small peptides that mimics the protein-rich environment of the stratum corneum of the tongue [34], and in BSA representing a more complex nitrogen source (**S4D Fig**). The better adaptation of isolate 101 compared to SC5314 in sensing and using GlcNAc was further evidenced by its resistance to GlcNAc-induced cell death (**S4E Fig**). GlcNAc serves as a signal of nutrient availability in *C.*

*albicans*, inducing entry into mitotic cell cycle and rapid cell growth. However, incubation in the presence of GlcNAc without other nutrients to support growth results in cell death due to the accumulation of intracellular ROS [35]. While we confirmed this phenomenon with strain SC5314 (**S4F Fig**), for which it was originally demonstrated [35], isolate 101 survived unabatedly under these conditions (**S4F Fig**), in line with its increased expression of oxidative stress resistance genes (**Fig 4D**). The metabolic profile of isolate 101 was not unique: eight other isolates that we tested for growth in GlcNAc and casamino acids displayed variable growth profiles, which did not correlate with the high- or low damage inducing capacity of the isolates in epithelial cells (**S4G Fig**). Altogether, these findings indicated that persistent growth of isolate 101 in the oral cavity was favoured not only by restricted filamentation and virulence, but also by metabolic adaption to the nutrient-poor environment in the stratified epithelium.

## Overexpression of NRG1 in SC5314 restrains its virulence potential *in vitro* and *in vivo*

Given the pronounced differences between isolates 101 and SC5314, we speculated that modulating the expression of virulence traits in isolate 101 would alter its phenotype and change its behaviour at the interface with the host. Given the link between damage induction and induction of inflammation in the host by highly virulent strains of *C. albicans* and because *ECE1* plays a prominent role in this process, we attempted to heterologously express *ECE1* of strain SC5314 in isolate 101. Despite choosing the strong *TDH3* promoter, we could not achieve *ECE1* expression levels that were high enough to elicit detectable levels of secreted peptides or to increase damage induction in keratinocytes (**S5A Fig**). The failure of achieving enhanced virulence in the commensal isolate 101 might be due to inadequate posttranslational processing of Ece1p required for generation of the peptide toxin known as Candidalysin [36] and the need for delivery of the toxin to the host cells via an invasion pocket at the hyphal tip [37]. The reduced filamentation of isolate 101 may not suffice those criteria.

We therefore turned our focus onto transcription factors that regulate *C. albicans* virulence more broadly and that were differentially regulated between the highly virulent SC5314 isolate and the low-virulent isolate 101. Among the most strongly regulated genes in isolate 101 was *NRG1* that encodes a repressor of the filamentation program. Given the filamentation phenotype of isolate 101 and the central role of filamentation *C. albicans* pathogenicity, we speculated that manipulating the expression of *NRG1* in isolate 101 might alter its phenotype and make it more virulent. Vice versa, overexpressing *NRG1* in SC5314 might have the opposite effect. Therefore, we made use of a SC5314-based isolate that conditionally overexpresses *NRG1* upon doxycycline (Dox) treatment [13]. Overexpression of *NRG1* (**Fig 5A**) did not affect the growth rate of the isolate (**S5B Fig**), but resulted in a defect in filamentation, both on YPD agar and in liquid medium. Dox-treated SN76$^{NRG1-OE}$ cells, but not cells of a Dox-treated control isolate, displayed a yeast/pseudophyphal morphology (**Fig 5B and 5C**). Moreover, *NRG1* overexpression was accompanied by reduced expression of diverse virulence genes such as *ECE1*, *ALS3* and *HWP1* (**Fig 5D**) and a reduced capacity to induce damage in TR146 keratinocytes (**Fig 5E**).

Based on these data, we wondered whether modulation of *NRG1* levels would also affect the behaviour of *C. albicans* cells *in vivo* in the oral mucosa. Pre-treatment of the isolate and supplementation of the drinking water of the mice with doxycycline had no effect on initial steps of the infection process (**Fig 5F**). However, the acute inflammatory response characterized by neutrophil recruitment to the site of infection in consequence of epithelial damage, was much reduced in mice infected with *C. albicans* strain overexpressing *NRG1* in comparison to controls as evidenced on histology sections (**Fig 5G**) and by enumeration of neutrophils in

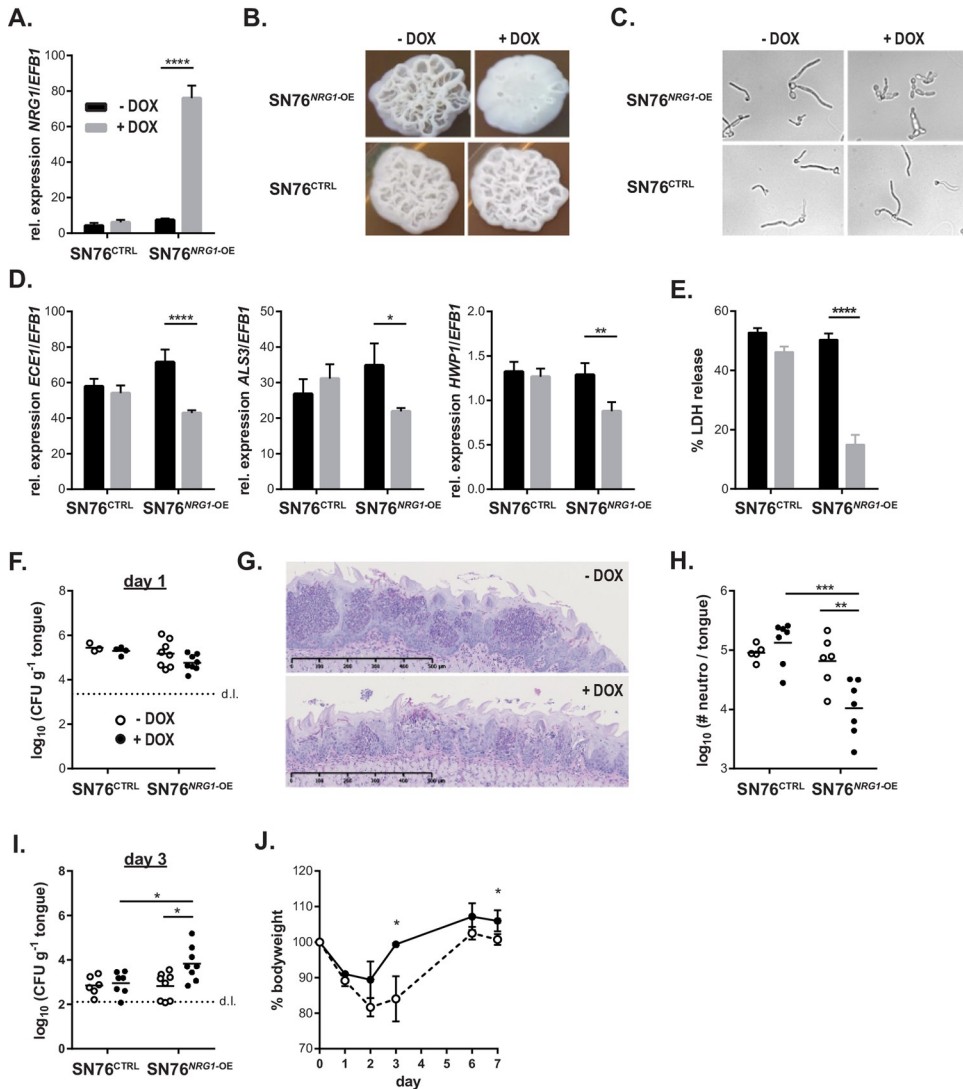

**Fig 5. Overexpression of NRG1 in SC5314 restrains its virulence potential *in vitro* and *in vivo*. A.** Monolayers of TR146 keratinocytes were infected with SN76$^{NRG1-OE}$ or SN76$^{CTRL}$ and treated with Dox or left untreated. RNA was isolated 24 hours post-infection and expression of *NRG1* was assessed by RT-qPCR. Bars are the mean+SD of 3 samples per conditions, representative of 2 independent experiments. **B-C.** Morphology of SN76$^{NRG1-OE}$ and SN76$^{CTRL}$ in presence or absence of Dox on YPD agar for 2 days (B) or in F12 medium for 8 hours (C). Data in C are representative of 2 independent experiments with 3–6 replicates per condition. The same results were obtained with two independent clones of SN76$^{NRG1-OE}$. **D.** Expression of the indicated genes were assessed as in (A). **E.** Monolayers of TR146 keratinocyte were infected with isolates SN76$^{NRG1-OE}$ or SN76$^{CTRL}$ in presence or absence of Dox and epithelial cell damage was assessed after 24 hours of infection by LDH release assay. Bars are the mean+SEM of 8 samples per condition pooled from 2 independent experiments. The same results were obtained with two independent clones of SN76$^{NRG1-OE}$. **F.–J.** C57BL/6 WT mice were infected sublingually with isolates SN76$^{NRG1-OE}$ or SN76$^{CTRL}$ and treated or not with Dox. Fungal burden was assessed after 1 day (**F**) or 3 days of infection (**I**). Tongue sections were stained with PAS (**G**) and neutrophils in the tongue were quantified by flow cytometry (**H**) on day 1 post-infection. Weight loss and re-gain relative to the pre-infection weight is shown in (**J**). In F, H and I, each symbol represents one animal, data are pooled from two independent experiments; in J each symbol is the mean±SD of 4 animals; images shown in G are representative of 2 independent experiments. Please see also **S4 Fig**.

infected tongues by flow cytometry (**Fig 5H**). The reduced inflammatory response of the host was accompanied by a delay in fungal clearance which manifested by elevated fungal loads on day 3 post-infection in case of the *NRG1*-overexpressing *C. albicans* strain in comparison to all

controls (**Fig 5I**). By day 7 however, mice had fully cleared the fungus and regained their initial weight (**Fig 5J**).

We also examined the consequences of modulating expression of another gene, *CRZ2*, which was also differentially expressed between isolates SC5314 and 101 and whose overexpression in SC5314 results in enhanced fitness during gastrointestinal colonization [13]. While overexpression of *CRZ2* in SC5314 induced a slight change in colony morphology and a reduction in epithelial damage induction *in vitro* (**S5C–S5E Fig**), we did not observe an effect of *CRZ2* overexpression on the inflammatory host response or a prolongation of oral colonization *in vivo* (**S5F–S5I Fig**).

## Reduced expression of NRG1 in isolate 101 increases its pathogenicity

Following from the results obtained with an *C. albicans* SC5314 derivative overexpressing *NRG1*, which revealed impaired virulence *in vitro* and *in vivo* in the OPC model, we wondered whether the opposite could be achieved in isolate 101, namely whether reducing *NRG1* expression levels in this isolate would be able and sufficient to increase its pathogenicity. Deletion of one or both alleles of *NRG1* in isolate 101 ($101^{nrg1\Delta/NRG1}$ and $101^{nrg1\Delta/\Delta}$, **Fig 6A**) entailed a dose-dependent increase in expression of determining virulence factors, including *ALS3*, *ECE1* and *HWP1* (**Fig 6B**). Expression of *ALS3* in $101^{nrg1\Delta/\Delta}$ reached levels even higher than those in wild type SC5314 (**Fig 6B**). The altered gene expression profile correlated with increased filamentation. Both, the heterozygous and the homozygous deletion mutants formed longer filaments than the parental isolate 101 when exposed to human keratinocytes (**Figs 6C and S6A**). Increased filamentation of the mutants was also observed on Spider agar and on YPD agar containing different carbon sources (**S6B and S6C Fig**) and this correlated with enhanced invasion properties (**S6C Fig**). Acquisition of these important virulence traits was however not sufficient for induction of detectable cellular damage in TR146 keratinocytes (**Fig 6D**).

Importantly, deletion of *NRG1* in isolate 101 resulted in enhanced virulence in the oral mucosa of experimentally infected mice *in vivo*. Deletion of a single copy of the gene was sufficient to elicit an enhanced inflammatory response characterized by increased expression of inflammatory cytokines and mediators and enhanced infiltration of neutrophils and monocytes into the colonized tongue as assessed by flow cytometry and on histology sections on day 1 post-infection, while colonization loads were comparable between $101^{nrg1\Delta/NRG1}$ and the parental isolate at this time point (**Fig 6E–6H**). Notably, the enhanced inflammatory response triggered by $101^{nrg1\Delta/NRG1}$ in the host impaired fungal colonization and resulted in a reduced fungal load on day 7 post-infection (**Fig 6I and 6J**).

Unfortunately, we were unable to extend our analysis to the homozygous deletion mutant. Isolate $101^{nrg1\Delta/\Delta}$ tended to form clumps in culture, which reduced the efficiency of the isolate to colonize the murine oral mucosa (**Fig 6E**). Clumping could not be prevented by filtering the isolate before applying it to the animals. The inequality in fungal load recovered from the tongue on day 1 post-infection (**S6D Fig**) prevented any further analyses of the $101^{nrg1\Delta/\Delta}$ mutant *in vivo*.

We therefore pursued a second independent approach to suppress *NRG1* expression in isolate 101 very efficiently using a Dox-dependent TET-off system. Notably, one *NRG1* allele was under the control of the TET promoter in the designed isolate, while the other was deleted ($101^{nrg1\Delta/pTET-NRG1}$) (**S7A and S7B Fig**). As little as 1 µg/ml of Dox resulted in a strong reduction of *NRG1* expression in isolate $101^{nrg1\Delta/pTET-NRG1}$ without overt clumping of the isolate (**Figs 7A and S7C**). Similar as with the *NRG1* deletion mutants, the reduction of *NRG1* transcript levels was accompanied by a dose-dependent increase in the expression of various

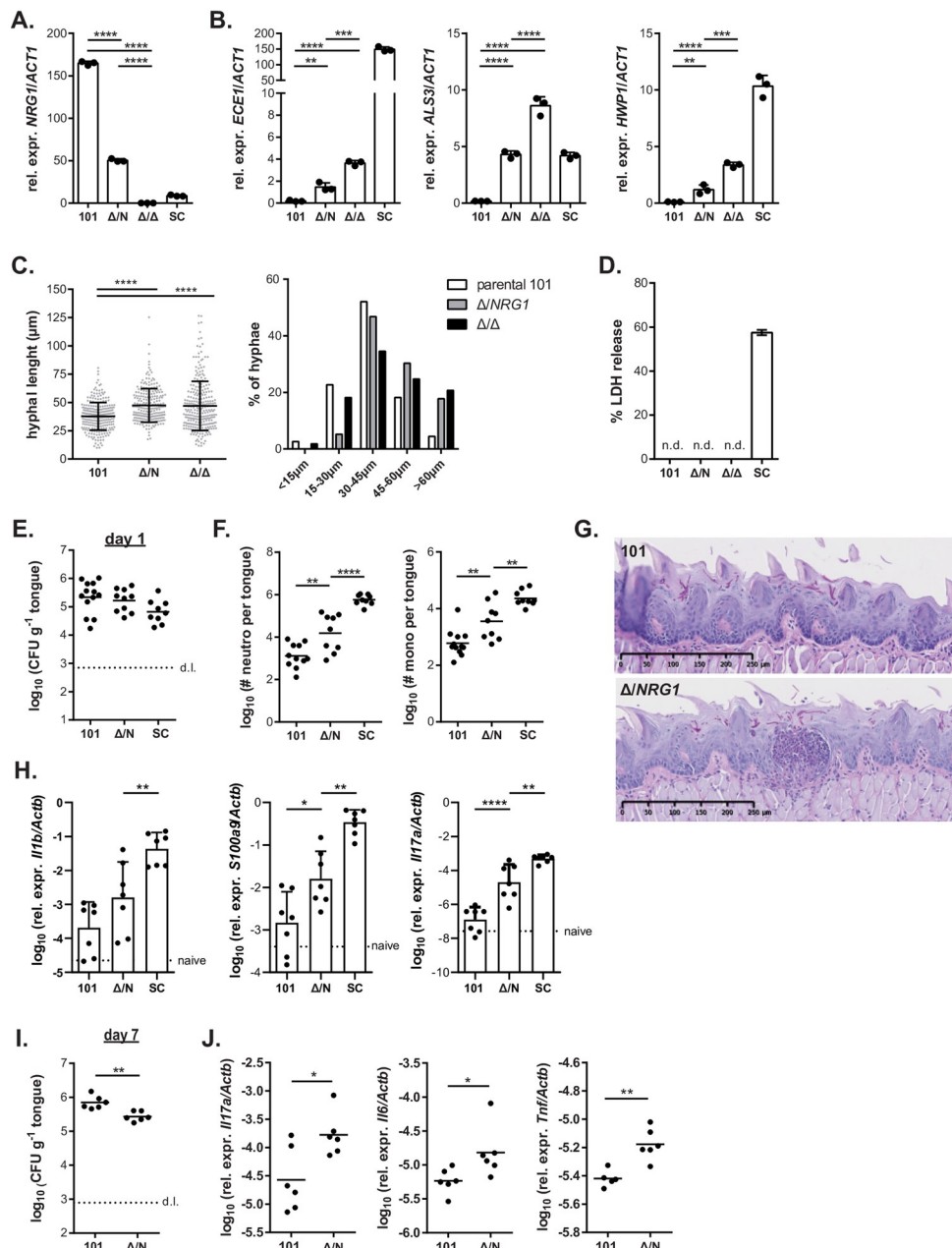

**Fig 6. Reduced expression of NRG1 in isolate 101 increases its pathogenicity. A.-D.** Monolayers of TR146 keratinocytes were infected with isolates $101^{nrg1\Delta/NRG1}$, $101^{nrg1\Delta/\Delta}$, the parental isolate 101 and strain SC5314. Expression of the indicated fungal genes was assessed by RT-qPCR after 24 hours of infection (**A, B**). Bars are the mean+SD of 3 samples per condition. Data are representative of 2 independent experiments. Filamentation was assessed after 3.5 hours of infection by measuring the length of individual filaments (**C**, left). Each dot represents an independent hyphal element; the mean ± SD is indicated. Fungal elements were grouped according to their length and the proportion of fungal elements in each group is shown (**C**, right). Data are representative of 3 independent experiments (only one experiment for $101^{nrg1\Delta/\Delta}$). Epithelial cell damage was assessed by LDH release assay at 24 hours post-infection (**D**). Bars are the mean+SEM of 10–22 samples per group pooled from 2–3 independent experiments. **E.-J.** C57BL/6 WT mice were infected sublingually with isolates $101^{nrg1\Delta/NRG1}$, the parental isolate 101 and SC5314. Fungal burden was assessed after 1 day (**E**) or 7 days of infection (**I**). Neutrophils and monocytes in the tongue were quantified by flow cytometry (**F**) and tongue sections were stained with PAS (**G**) on day 1 post-infection. The indicated host transcripts in the tongue were quantified by RT-qPCR on day 1 (**H**) or day 7 post-infection (**J**). In E, F, H, I and J, each symbol represents one animal; the mean is indicated. Data in E, F, H, I and J are pooled from 2–3 independent experiments. Please see also **S5 Fig**.

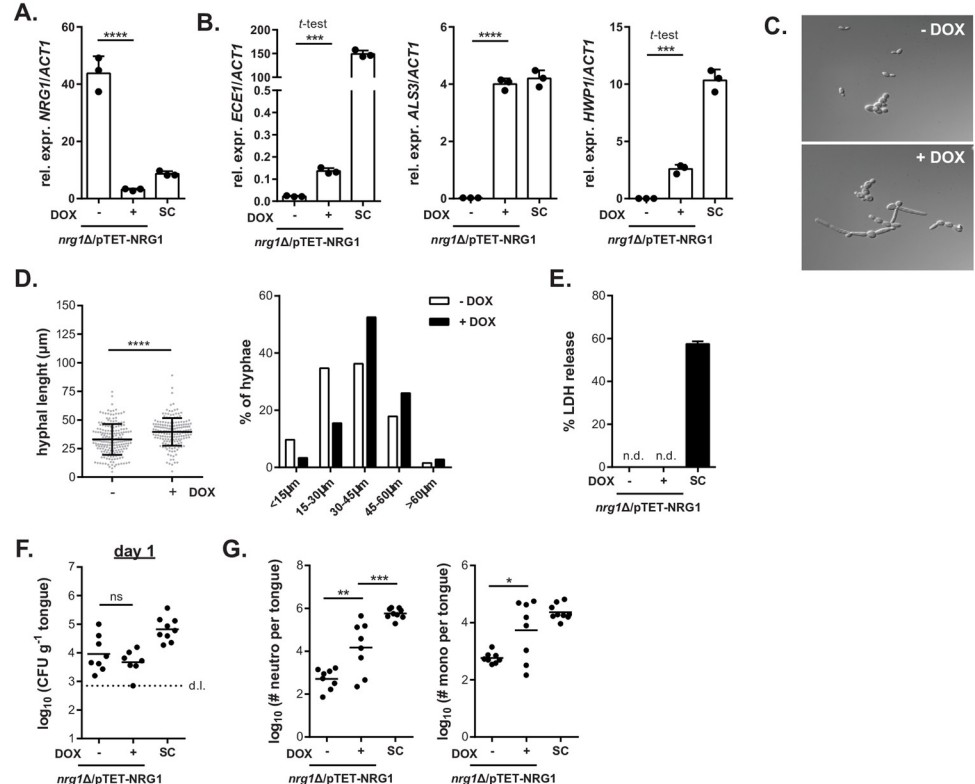

**Fig 7. Suppression of NRG1 expression in isolate 101 via a TET-off strategy drives fungal pathogenicity. A.-B.**
Monolayers of TR146 keratinocytes were infected with isolates 101$^{nrg1\Delta/pTET-NRG1}$ in presence or absence of 10 μg/ml
Dox and with SC5314. Expression of the indicated fungal genes was assessed by RT-qPCR after 24 hours of infection.
Bars are the mean+SD of 3 samples per condition. Data are representative of 2 independent experiments. **C.** 101$^{nrg1\Delta/pTET-NRG1}$ yeast cells were grown for 4 hours at 37˚C in YPD with or without addition of 10 μg/ml Dox. **D.** Monolayers
of TR146 keratinocytes were infected with isolate 101$^{nrg1\Delta/pTET-NRG1}$ in presence or absence of 10 μg/ml Dox.
Filamentation was assessed after 3.5 hours of infection by measuring the length of individual filaments (left). Each dot
represents an independent hyphal element; the mean ± SD is indicated. Fungal elements were grouped according to
their length and the proportion of fungal elements in each group is shown (right). **E.** Epithelial cell damage was
assessed by LDH release assay at 24 hours post-infection. Bars are the mean+SEM of 10–16 samples per group pooled
from 2 independent experiments. **F.-G.** C57BL/6 WT mice were infected sublingually with isolate 101$^{nrg1\Delta/pTET-NRG1}$
or SC5314 for 1 day. Fungal burden was assessed in **F**. Neutrophils and monocytes in the tongue were quantified by
flow cytometry in **G**. Each symbol represents one animal; the mean is indicated. Data are pooled from 2 independent
experiments. Please see also **S6 Fig**.

virulence factors (**Figs 7B** and **S7C**). Dox -induced suppression of *NRG1* expression also
resulted in enhanced filamentation, both under YPD culture conditions at 37˚C without addi-
tion of FCS or any other known hyphae-inducing agents (**Fig 7C**), and after exposure to
TR146 keratinocytes (**Fig 7D** and **S7D**). Importantly, these phenotypic and functional changes
were accompanied by enhanced pathogenicity *in vivo* as indicated by the strong inflammatory
response induced by isolate 101$^{nrg1\Delta/pTET-NRG1}$ in the oral mucosa upon Dox treatment (**Fig 7F
and 7G**). Together, these results demonstrated that *NRG1* acts as a central regulator of *C. albi-
cans* pathogenicity in the oral cavity and that suppression of high *NRG1* expression levels in
the commensal stain 101 is sufficient to increase its virulence and to alter the fungus-host
interplay at the epithelial barrier.

## Discussion

The oral cavity is an important niche for *C. albicans* colonization [17]. It also represents a site where infections caused by *C. albicans* frequently manifest [18]. Our understanding of the fungal cues that allow *C. albicans* to persist in this niche and to be tolerated by the host without causing disease, remains incomplete. To fill this gap in knowledge, we employed a model of oral colonization in mice and examine requirements for *C. albicans* commensalism using a prototypical commensal isolate as an example.

Transcript profiling of *C. albicans* isolate 101 in the oral tissue and functional studies with the isolate in host-involving and host-free conditions revealed a pronounced reduction in multiple virulence traits, including filamentation, invasion and damage induction. Based on this information, we attempted to increase virulence of isolate 101. Overexpression of specific virulence genes, such as *ECE1*, was not sufficient to induce a gain in pathogenicity in isolate 101, possibly consistent with the recent demonstration that the full damage potential of *C. albicans* depends not only on strong Ece1 expression but also coordinated delivery of fully processed and secreted Candidalysin in invasion pocket created by invasive hyphae [37]. Of note, we expressed the entire *ECE1* gene originating from SC5314 under the strong *TDH3* promoter, including the sequences flanking peptide 3 for optimal Candidalysin processing, to exclude suboptimal processing of Ece1p as recently reported [38].

Strengthening the hyphal program in isolate 101 by suppressing *NRG1* expression resulted in major changes in its behaviour in the oral cavity. Forced induction of filamentation in the normally avirulent isolate 101 was sufficient to elicit an inflammatory response in the colonized host and to impair colonization, indicating that restricted filamentation is a major determinant of commensalism in the oral cavity. This is reminiscent of what was described for other commensal niches of *C. albicans* [15,39]. Despite the parallels in the role of filamentation in commensalism in various mucosal epithelia, there are also important differences between different sites. As such, SC5314$^{CRZ2-OE}$ acquires enhanced fitness during gut colonization [13] but not during OPC suggesting that commensalism in the gastrointestinal tract and in the oral cavity integrates different fungal effectors. Morphotype switching drives *C. albicans* pathogenicity also during systemic infection [40]. Of interest, rapid yeast proliferation in internal organs can compensate for the lack of filamentation, as it was shown recently with an *EED1* deficient mutant [41].

*NRG1* is a transcriptional repressor of hyphae-specific genes and of genes required for the morphogenic switch [42]. *NRG1* transcription is downregulated under conditions promoting filamentation. Forced constitutive expression of *NRG1* in SC5314 established the need of morphotype switching for pathogenicity of *C. albicans* during disseminated candidiasis [40], while enforced filamentation due to constitutive deletion of *NRG1* lowered intestinal colonization of *C. albicans* [43]. A limited degree of filamentation is however beneficial for commensalism in the gut: it has recently been shown to drive an IgA response targeting fungal hyphae, which in turn improves competitive fitness of *C. albicans* and promotes homeostasis [44]. The role of *NRG1* as a driver of commensalism extends beyond the species of *C. albicans*: reduced virulence of *C. dubliniensis* was linked to the inability of this species to modulate *NRG1* expression in response to environmental signals that promote filamentation in prototypic isolates of *C. albicans* [45].

Determining the genetic basis for the commensal phenotype of *C. albicans* isolate 101 represents a significant challenge due to the intrinsically high diversity in *C. albicans*' population genetic structure [3], consisting of numerous sequence and, higher order genomic differences [46,47]. The situation is further complicated by circuit diversification in regulatory networks that exist between isolates [48] and the fragility of such networks in the oral mucosa [49]. This

large complexity prevented us from yet identifying any specific genetic determinants underlying the commensal phenotype of isolate 101. No significant sequence differences were identified in the promoter region of *NRG1* between isolate 101 and SC5314. Replacing an allele of an upstream transcriptional regulator of *NRG1* (*BRG1*), which we found to be truncated in isolate 101, with a full-length copy was not sufficient to reduce *NRG1* expression and alter the phenotype of the isolate. A genetic determinant of *C. albicans* commensalism was recently identified in the coding sequence of *ECE1*, which affects processing of the damage causing peptide toxin and may thereby contribute to differential pathogenicity among clinical isolates [38]. While this may explain, at least in part, the low virulence of isolate 529L in the OPC model [7,50], it does not so for isolate 101, which displays the SC5314 isoform at the relevant *ECE1* P2-P3 junction. Establishing the genetic determinant, or rather the combination of determinants, responsible for the commensal lifestyle of isolate 101 remains a challenging task for the future.

The transcriptome of *C. albicans* isolate 101 in the oral tissue revealed a metabolic signature adapted for nutrient acquisition in this specific niche. In experimentally infected mice, commensal isolates are confined to the stratum corneum of the tongue [7], the outermost layer of the stratified epithelium composed of metabolically inactive corneocytes that form a tight and water impermeable barrier limiting nutrient availability [51]. The absence of tissue damage by commensal isolates further limits access to nutrients inside host cells. The high expression of genes encoding proteases, oligopeptide transporters and amino acid catabolic enzymes indicates customized nutrient acquisition mechanisms being active in this protein-rich tissue compartment. Among the amino acid catabolic enzymes whose genes are especially highly expressed in isolate 101 were those involved in catabolism of Glycine and Glutamine, which are highly enriched in structural proteins of the cornified envelope of epithelium [34].

The selective localization to the stratum corneum may thus not simply be the consequence of impaired damage induction by isolate 101 but also reflect metabolic specialization to this niche. This is further supported by the expression of specific subsets of *SAP* genes with *SAP1* and *SAP2*, which mediate initial invasion of the epithelium [30], being highly expressed by isolate 101 in contrast to *SAP6* and *SAP8*, which are associated with penetration of *C. albicans* to deeper epithelial layers [30]. The restricted localization of commensal *C. albicans* in the stratum corneum may promote fungal persistence in several ways: Exclusion from deeper layers of the epithelium limits induction and release of inflammatory mediators from metabolically active epithelial cells located in the stratum spinosum. It also protects the fungus from exposure to antifungal mechanisms of the epithelium such as antimicrobial peptides, which are lower in concentration towards the outside of the epithelium because the cornified envelope, a water-proof amalgam of crosslinked proteins and hydrophobic lipids, hardly provides access to water-soluble positively charged antimicrobial peptides. Enhanced resistance to host defences represents an additional mechanism of immune evasion that allows the fungus to survive homeostatic immunity, which the host raises to counteract fungal (over)growth [21]. *C. albicans* isolate 101 did even resist the strong inflammatory host response to which it was exposed during co-infection with the high-virulent SC5314, which suggests the acquisition of several factors inducing host resistance by isolate 101. Among these, resistance to oxidative stress prevailing in 101 provides an advantage during exposure to extracellular ROS, e.g. from neutrophils. An enhanced oxidative stress response also contributes to protection from GlcNAc-induced cell death by counteracting the accumulation of intracellular ROS [35]. Resistance to oxidative stress may be further enhanced by GlcNAc and amino acids [52], which are favourably metabolized by isolate 101.

That the metabolic program of *C. albicans* can be at least as important as its virulence for its behaviour in a specific host niche has recently been demonstrated by the Jacobsen lab. Using a filamentation-deficient mutant strain they demonstrated that metabolic adaptation, rather

than filamentation, was important for gut pathogenicity [41]. In case of isolate 101, the metabolic program provides this strain with an advantage for thriving in the nutrient-poor cornified layer of the tongue epithelium and may thereby favour its low pathogenicity. Here we observed that *NRG1* expression can modulate host-pathogen interactions, however the precise role of *NRG1* in the mode of regulation of isolate 101's metabolic profile remains still to be precisely dissected.

Isolate 101 is a prototypical commensal isolate, which we studied in its natural niche. Based on its low damage inducing capacity, it is representative of the majority of commensal isolates [7]. Similarly, with respect to its nutrient preferences, isolate 101 is not an outlier within the species of *C. albicans*. Importantly, however, every isolate is unique in how it combines different properties, as it becomes clear when comparing a larger number of properties across isolates [7]. While individual *C. albicans* isolates display multiple phenotypic differences when analysing them in isolation, a strain's lifestyle at the host interface is the result of the combination of all traits and in dependence of the niche environment. This has also been highlighted by a recent study of the Ene, Koh and Hohl labs, which in a joint effort demonstrated that multiparametric strain-intrinsic factors rather than lab- or animal facility-dependent parameters account for the commensal lifestyle of isolates [53]. Moreover, colonization of the mucosal niche by a particular strain is not readily predictable from the *in vitro* analysis of individual features of that strain [49].

Together, our work sheds new light on fungal attributes of *C. albicans* commensalism in the oral cavity. Improved understanding of how the fungus can prevent pathogenicity in various host niches paves the way for novel therapeutic approaches that aim not at eradicating the fungus but rather counteracting its virulence. This strategy has been successfully approached with antagonizing bacteria [54,55] and compounds targeting *C. albicans* hyphae and biofilm formation via *TUP1* have been developed [56]. Based on this concept new opportunities may also open up for the NDV-3A vaccine, which can shift the hyphae/yeast balance by inducing ALS3-specific IgA [44]. Targeting fungal virulence instead of the fungus' ability to thrive on colonized surfaces will further help limiting the rise in antifungal resistance.

## Methods

### Ethics statement

All mouse experiments in this study were conducted in strict accordance with the guidelines of the Swiss Animals Protection Law and were performed under the protocols approved by the Veterinary office of the Canton of Zürich, Switzerland (license number 183/2015 and 166/2018). All efforts were made to minimize suffering and ensure the highest ethical and humane standards according to the 3R principles.

### Fungal strains

Strains used in this study are listed in Table 1. All strains were maintained on YPD agar for short term and in glycerol-supplemented medium at -80˚C for long term storage. Cultures were inoculated at $OD_{600}$ = 0.1 in YPD medium (using a pre-culture to adjust the $OD_{600}$) and grown at 30˚C and 180 rpm for 15–18 hours. For Dox inducible strains, the cultures were supplemented with 50 μg/ml (in case of SC5314$^{NRG1-OE}$) or 10 μg/ml Dox (in case of 101 $^{nrg1\Delta/}$ $^{pTET-NRG1}$, unless stated otherwise). At the end of the culture period, yeast cells were washed in PBS and their concentration was determined by spectrophotometry, whereby 1 $OD_{600}$ = $10^7$ yeast cells.

**Table 1. C. albicans strains used in this study.**

| Strain | Collection # | genotype | Parental | reference |
|---|---|---|---|---|
| SC5314 | CA1 | Clinical isolate | | [57] |
| SC5314$^{GFP}$ | CA2 | pACT1-GFP | CAI4 | [22] |
| 101 | CA117 / DSY4709 | Clinical isolate | | [7] |
| 101$^{mCherry}$ | DSY4718 / CA124 | *ADH1*::pADH-Cherry-*SAT1* | 101 | [21] |
| SN76 | CEC4665 / CA210 | SN76 *ADH1/adh1*::P$_{TDH3}$-carTA::*SAT1 arg4Δ*::*CaARG4 his1Δ*::*hisG/HIS1 RPS1/RPS1*::CIp10 | CEC4642 | this study |
| SN76$^{NRG1-OE}$ | CEC6039 / CA227 | SN76 *ADH1/adh1*::P$_{TDH3}$-carTA::*SAT1 arg4Δ/CaARG4 his1Δ*::*hisG/HIS1 RPS1/RPS1*::CIp10-P$_{TET}$-*NRG1* | CEC4642 | this study |
| SC5314$^{CTRL2}$ | CEC4442 / CA222 | *ura3Δ*::*λimm434/ura3Δ*::*λimm434, RPS1/RPS1*::CIp10-P$_{TET}$-*URA3*-GTW, *iro1Δ*::*λimm434/iro1Δ*::*λimm434, arg4Δ/ARG4 ADH1/adh1*::P$_{TDH3}$-carTA::*SAT1* Ca21chr4_C_albicans_SC5314:473390 to 476401Δ::P$_{TDH3}$-BFP-*HIS1* | CEC3783 | [13] |
| SC5314$^{CRZ2-OE}$ | CEC4439 / CA221 | *ura3Δ*::*λimm434/ura3Δ*::*λimm434, RPS1/RPS1*::CIp10-P$_{TET}$-*CRZ2-URA3*-GTW, *iro1Δ*::*λimm434/iro1Δ*::*λimm434, his1Δ/HIS1, ADH1/adh1*::P$_{TDH3}$-carTA::*SAT1*, Ca21chr4_C_albicans_SC5314:473390 to 476401Δ::P$_{TDH3}$-GFP-*ARG4* | CEC3781 | [13] |
| | DSY5577 | *nrg1Δ*::*FRT1-SAT1-FRT1/NRG1* | DSY4709 | this study |
| 101$^{nrg1Δ/NRG1}$ | DSY5592 / CA245 | *nrg1Δ*::*FRT1/NRG1* | DSY5577 | this study |
| 101$^{nrg1Δ/Δ}$ | DSY5599 / CA246 | *nrg1Δ*::*FRT1/nrg1Δ*::*FRT1-SAT1-FRT1* | DSY5592 | this study |
| | DSY5617 | *nrg1Δ*::*FRT1/nrg1Δ*::*FRT1* | DSY5592 | this study |
| 101$^{nrg1Δ/pTET-NRG1}$ | DSY5625 / CA252 | *nrg1Δ*::*FRT1/pTET-O7-NRG1*::*NAT1 ADH1*::pDS2142 | DSY5617 | this study |
| DSY5622 | DSY5622 | *ura3Δ*::*λimm434/ura3Δ*::*λimm434, iro1Δ*::*λimm434/iro1Δ*::*λimm434, arg4Δ/ARG4, his1Δ*::*hisG/HIS1, RPS1/RPS1*::CIp-PTET-SP-gLUC59 | CEC161 | this study |
| DSY5624 | DSY5624 | *ura3Δ*::*λimm434/ura3Δ*::*λimm434, iro1Δ*::*λimm434/iro1Δ*::*λimm434, arg4Δ/ARG4, his1Δ*::*hisG/HIS1, RPS1/RPS1*::CIp-PTET-SP-gLUC59, ADH1*::pDS2142 | DSY5622 | this study |
| 101$^{TDH3-ECE1}$ | DSY5209 | *TDH3*::pDS2037 | 101 | this study |
| 529L | CA115 | Clinical isolate | | [50] |
| Cag | CA116 | Clinical isolate | | [7] |
| CEC3605 | CA131 | Clinical isolate | | [58] |
| CEC3609 | | Clinical isolate | | [58] |
| CEC3672 | | Clinical isolate | | [59] |
| CEC3621 | | Clinical isolate | | [58] |
| CEC3678 | | Clinical isolate | | [59] |
| CEC3617 | | Clinical isolate | | [58] |

## Generation of NRG1 overexpressing mutants

*NRG1* was PCR amplified from *C. albicans* SC5314 genomic DNA using primers 23_G09_FP and 23_G09_RP. PCR product was mixed with the donor plasmid pDONR207 (Invitrogen), and subjected to a recombination reaction with Invitrogen Gateway BP Clonase, as described [60], to yield Entry vectors. The recombination mixes were transformed into *E. coli* Top10 and one transformant per ORF was selected for further study. The cloned ORFs were sequenced to ascertain that no mutations were introduced during PCR amplification. The Entry plasmids were used in a Gateway LR reaction together with the CIp10-P$_{TET}$-GTW vector [61]. The recombination mixes were transformed into *E. coli* Top10 and one transformant was used for plasmid preparation. BsrGI digestion was used to verify the clones. The expression plasmids

were digested by StuI prior to transformation into *C. albicans* strain CEC4642. Transformants were selected for prototrophy and proper integration at the *RPS1* locus verified by PCR using primers CIpUL and CIpUR, as described [60].

## Generation of NRG1 deletion mutants

*NRG1* was deleted in isolate 101 using a *SAT1*-recyclable system [62]. Briefly, two primers containing 70-pb of sequences flanking the *NRG1* ORF and 20-bp of sequences flanking the *SAT1*-flipper cassette were used in a PCR with pSFS2A [62] as a template. This step was performed with Phusion DNA Polymerase (New England Biolabs, Ipswich, MA, USA) in the presence of 1M betaine. The PCR fragment was used to transform the isolate 101 by electroporation [63] and under selection with nourseothricin (400 μg/ml; Werner BioAgents, Jena, Germany). After PCR verification of the first allele deletion by primers pSFS2A Dwn Chk et NRG1_Kpn (isolate DSY5577), the *SAT1* marker was recycled as described [62] to yield DSY5592. This last isolate was used for second allele inactivation resulting in DSY5599.

## Generation of Tet-off mutant

To generate *NRG1* downregulation in the isolate 101, the tetracycline reversed activator (CartTA) in pNIMX [61] was replaced after XbaI-MluI digestion by the tetracycline activator reported in Bijlani et al. [64] to result in pDS2136. The CatTA gene was prepared as a synthetic fragment (Eurofins Genomics GmbH, Ebersberg, Germany). The hygromycin resistance gene (HygR) was amplified with primers Hygro-TEF2-Nde et Hygro-ACT1-NdeI using pMY70 [65] (see Table 2) and replaced *SAT1* as a NdeI fragment in pDS2136 to produce pDS2142. This plasmid was linearized by KpnI/SacII for integration at the *ADH1* locus in isolate DSY5592 to yield DSY5617 under selection with hygromycin (600 μg/ml; Mediatech, Manassas, VA, USA).

To construct a tetracycline-dependent promoter system, a PCR fusion strategy was chosen. A first fragment containing a tetracycline operator (pTETO7-OP4) was generated with primers Tetp-5 and Tetp-3 using pNIM1 [61] as template. A second PCR fragment containing the *NAT1* dominant marker was generated with overlapping ends using pJK795 [66] as template. Both fragments were used in a PCR fusion with primers of 70-bp overlapping *NRG1* matching regions upstream and downstream of the ORF (NAT1-NRG1 and TetP-3-NRG1). This last step was carried out with Phusion DNA Polymerase (New England Biolabs, Ipswich, MA, USA) in the presence of 1M betaine. The final PCR fragment was precipitated with 70% EtOH and transformed in DSY5617 (*NRG1/nrg1*Δ::FRT *ADH1*::pDS2142) to yield DSY5625 which contains *NRG1* under the control of pTETO7-OP4.

To test the Tet-off system, a strain with a luciferase reporter system was created by introducing CIp-PTET-SP-gLUC59 after StuI digestion in CEC161 [61], resulting in isolate DSY5622. Plasmid pDS2142 was next introduced in this isolate to yield DSY5624.

## Luciferase assay

Isolate DSY5624 was cultured overnight in YEPD and grown to log phase for 4 hr in the same medium (5 ml) containing different Dox concentrations. After incubation, cells were washed and resuspended in 200 μl R-luc buffer (0.5 M NaCl, 0.1 M $Na_2HPO_4$ pH 7.0, 1 mM EDTA). Eighty μl of each suspension was pipetted in duplicates into black half area 96-well plate. Water-soluble coelenterazine (catalog # 3032, Nanolight Technology, Pinetop, AZ, USA) was resuspended in 5 mL PBS (phosphate buffered saline: 137 mM NaCl, 2.7 mM KCl, 4.3 mM $Na_2HPO_4$, 1.47 mM $KH_2PO_4$) and kept in the dark. Plates were placed in a FLUOstar Omega instrument (BMG Labtech, Champigny/Marne, France). The instrument was kept at a

**Table 2. Primers used for generating C. albicans mutants.**

| primer | Sequence (5' -> 3') |
|---|---|
| KO primer fwd NRG1 | AAACATCGTTATCCTGTTTCTCATCTCAAAATTTTTCCCTGCTAGTTTCATTAAGAATCAAACAATCATTCTCGAGGAAGTTCCTATACT |
| KO primer rev NRG1 | AAAAAAAACTAAACCCAAGCAATTAACCATCCAAATTTAACCCGTTTTATAATACAATTTTGACCACATGTGTGGAATTGTGAGCGGATA |
| Psfs2a Dwn Chk | ACAGCGATGTACTGGTACTG |
| NRG1_Kpn | CCAACTAGGGTACCATCATTATAATTAACCCCTC |
| Hygro-TEF2-Nde | CGCGAAACATATGTATAGTGCTTGCTGTTCGA |
| Hygro-ACT1-NdeI | AAGCGCCATATGATTTTATGATGGAATGAATG |
| Tetp-5 | ACTGCTGTCGATTCGATACTAATTAACCCTCACTAAAGGGAACAAAAG |
| Tetp-3 | GATATCGCCATTGTAAATTATTTATA |
| NAT1-3 | CCCTTTAGTGAGGGTTAATTAGTATCGAATCGACAGCAGTATAGCGAC |
| NAT1-5 | CCCTCCTTGACAGTCTTGACGTGCGC |
| NAT1-NRG1 | GAATCTGAAACAGGTATTATATAAATAACAATAAACATCGTTATCCTGTTTCTCATCTC AAAATTTTTCCTGTCGCCCGTACATTTAG |
| TetP-3-NRG1 | AATATAAATAGTCGACAAAGAGTTTCATTAAGAATCAAACAATCATTATGCTTTATCAAC AATCATATCCAATAACAAATAAGTTATTAA |
| HWP1_Kpn | AAAGTGGTACCAAAGCTATGATAAATGTTGATT |
| HWP1_XhoI | ATAATCTCGAGTTGACGAAACTAAAAGCGAGTG |
| ECE1_XhoI | TCCACTCGAGAAAATGAAATTCTCCAAAATT |
| ECE1_MluI | AGAGACGCGTAAGTAAAATATAGGTAATATAAAC |
| TDH3-XHOI | CGCGAACTCGAGTGTTAATTAATTTGATTGTAAAG |
| TDH3-KpnI | GGAACGGTACCATACAGTATTCAGTATGAT |
| 23_G09_FP | GGGGACAAGTTTGTACAAAAAAGCAGGCTtgATGCTTTATCAACAATCATATCCAATAACA |
| 23_G09_RP | GGGGACCACTTTGTACAAGAAAGCTGGGTcTACTAGGCTCTTGGTGTTGTATTTTGTTCC |
| CIpUL | ATACTACTGAAATTTCCTGACTTTC |
| CIpRL | ATTACTATTTACAATCAAAGGTGGTC |

temperature of 30˚C. After shaking the plate for 3 s and injection of 20 μl coelenterazine, reading of luminescence was started and continued with intervals of 1.5 s for 50 s.

## Generation of ECE1 overexpression mutant

*ECE1* was overexpressed in isolate 101 using the *TDH3* promoter. The promoter was amplified by PCR using genomic DNA from SC5314 with primer pairs TDH3-XHOI/TDH3-KpnI containing KpnI and XhoI sites. After digestion of the PCR fragments by KpnI and XhoI, the promoter was cloned in the plasmid pDS2032 to yield pDS2037. Plasmid pDS2032 contains *ECE1* from SC5314 under the control of the *HWP1* promoter in the backbone of pDS1904 [67], a derivative of CIp10 with a *SAT1* dominant marker. The *HWP1* promoter and *ECE1* were cloned sequentially from PCR fragments obtained with primers pairs HWP1_Kpn/ HWP1_XhoI and ECE1_XhoI/ECE1_MluI. Insertion of the KpnI-XhoI and XhoI-MluI digested PCR fragments yielded pDS2032. Plasmid pDS2037 was digested by MlscI and SmaI for introduction at the *TDH3* loci of isolate 101.

## Keratinocyte cell line and cell culture conditions

The oral keratinocyte cell line TR146 [68] was grown in DMEM medium supplemented with 10% FCS, Penicillin and Streptomycin at 37˚C and 5% $CO_2$. For experiments, cells were seeded at $2 \times 10^5$ cells/well in 6-well tissue culture plates or at $2 \times 10^4$ cells/well in 96-well tissue culture

plates, respectively, and grown to confluent monolayers for 2 days prior to infection as described below for the individual assays. 1 day prior to the experiment, the DMEM medium was replaced by F12 medium (Hams's Nutrient Mixture F12 medium (Sigma) supplemented with L-Glutamine, Penicillin, Streptomycin, and 1% FCS).

### Reconstituted human epidermis

Reconstructed human epidermis (RHE) models aged for 17 days were purchased from Episkin and cultured in 6-well tissue culture plates with 2 ml T-skin medium per well at 37°C and 5% $CO_2$. One day prior to infection, the models were transferred from T-skin medium to F12 medium containing 1% FCS and infected with 4 x $10^6$ yeast cells. The models were then carefully swayed using sterile forceps to distribute the liquid in the insert. After incubation for 24h at 37°C and 5% $CO_2$, the supernatant was collected for cell damage quantification by LDH assay. The models were cut in half and used for RNA isolation and histology, respectively. For histology, the support of the skin reconstructs was detached from the insert with a scalpel prior to embedding the tissue in 4% paraformaldehyde.

### *C. albicans* growth curve experiments

*C. albicans* was inoculated at $OD_{600} = 0.1$ in 200 μl YPD (10 g yeast extract, 20 g bacto peptone and 20 g D-glucose per 1L $H_2O$), YPGlcNAc (10 g yeast extract, 20 g bacto peptone and 20 g GlcNAc per 1L $H_2O$), YPCasamino acids (10 g yeast extract, 20 g bacto peptone and 20 g casamino acids per 1L $H_2O$) or YPBSA (10 g yeast extract, 20 g bacto peptone and 20 g BSA per 1L $H_2O$) in a 96well plate and grown in an Infinite M200 PRO plate reader (Tecan) for 24 hours at 30°C with orbital shaking (3 mm amplitude) and $OD_{600}$ measurements every 10 minutes that were preceded by a 10 s linear shaking pulse (1 mm amplitude).

### Morphological analyses

For assessing filamentation on spider agar (10 g mannitol, 10 g nutrient broth, 2 g $K_2HPO_4$ and 13.5 g agar per 1L $H_2O$), 3 μl of a yeast cell suspension at $10^5$ cells/ml were spotted and plates were incubated for 5 days at 30°C.

For assessing colony shape of SC5314[NRG1-OE] and SC5314[crz2-OE], 3 μl of a yeast cell suspension at $10^6$ cells/ml was spotted onto YPD agar containing 10% FCS and supplemented with 50 μg/ml Dox and plates were incubated for 2 days at 37°C.

For assessing colony shape of isolates 101[nrg1Δ/NRG1] and 101[nrg1Δ/Δ], 3 μl of a yeast cell suspension at $10^5$ cells/ml was spotted onto Spider agar, YPD agar (10 g yeast extract, 20 g bacto peptone, 20 g D-glucose and 20 g agar per 1L $H_2O$) or YPGlcNAc agar (10 g yeast extract, 20 g bacto peptone, 20 g GlcNAc and 20 g agar per 1L $H_2O$) and plates were incubated for 5 days at 30°C.

For measuring filament length, $5x10^4$ yeast cells were seeded in F12 medium (see above) into 24-well tissue culture plates containing or not containing monolayers of TR146 cells. For analysis of filamentation of 101[nrg1Δ/pTET-NRG1], 10 μg/ml Dox was added. After 3.5–4 hours of incubation at 37°C and 5% $CO_2$, cells were fixed in 4% PFA for 30 min at 4°C and washed with PBS. Images were acquired using a Zeiss Axio Observer Z1 inverted phase contrast fluorescence microscope. 4 fields per well were imaged. The length of individual non-aggregated filaments was measured using the ImageJ software.

### Invasion assays

For agar invasion, 5 μl of a yeast cell suspension at $10^7$ cells/ml were spotted on YNB/BSA agar (1.7 g yeast nitrogen base w/o amino acids and ammonium sulphate, 5 g ammonium sulphate,

20 g agar, 2 g BSA per 1L $H_2O$) and incubated for 4 days at 37˚C and 5% $CO_2$. Invasion depth was assessed after cutting the agar with a scalpel through the colonies. % invasion was calculated as the proportion of colony invasion relative to the entire agar thickness.

For assessing invasion of *C. albicans* into TR146 keratinocytes, monolayers of TR146 cells on sterile glass slides in 24-well tissue culture plates prepared as described above were infected with 5 x $10^4$ yeast cells per well. After 3 hours of incubation at 37˚C and 5% $CO_2$, cells were fixed in 4% PFA for 15–30 minutes and subjected to a two-step immunofluorescence staining protocol adapted from Wächtler et al. [69]. Extracellularly fungal parts were stained with an anti-*Candida* antibody (Meridian Life Science B65411R) followed by AF647-conjugated goat anti-rabbit IgG (Jackson Immunoresearch). After permeabilization with 0,5% Triton X-100, actin was stained with FITC-labelled phalloidin (Sigma) to visualize epithelial cells and fungal elements were stained with calcofluor white (Sigma). Images were acquired with a Leica SP8 inverse confocal microscope. The % of invasive hyphae was calculated relative to the total number of hyphae per imaged field. The number of total hyphae per field was determined by counting the buds (heads) from which the hyphae have emerged. Invasive hyphae were identified as those with clearly blue stretches. A total of 32 fields were quantified per sample.

## Phospholipase activity

5 µl of a yeast cell suspension at $10^7$ cells/ml were spotted on Prices' original egg yolk agar (4% glucose, 1% peptone, 1 M NaCl, 2,5 mM $CaCl_2$, 17 g agar per 1L $H_2O$ and addition of 2,5% eggyolk (Sigma) after autoclaving) and incubated for 4 days at 37˚C and 5% $CO_2$. Phospholipase activity was determined as the ratio of the diameter of the colony relative to the diameter of the precipitation zone forming around the colony.

## Cell damage assay

Damage induction by *C. albicans* in TR146 cells was performed as described [70]. Briefly, monolayers of TR146 cells in 96-well tissue culture plates prepared as described above were infected with2 x $10^4$ yeast cells per well and incubated for 24 hours at 37˚C and 5% $CO_2$. Control wells were incubated with medium only or with 1% Triton-X-100 to determine 100% damage. LDH release into the supernatant was quantified with the LDH cytotoxicity kit (Roche) according to the manufacturer's instructions. For assessing LDH concentrations in the supernatant of reconstructed human epidermis, a standard curve was generated with recombinant LDH enzyme.

## Cytokine quantification by ELISA

IL-6, IL-8, and GM-CSF levels in the supernatant of infected TR146 monolayers and RHE models were quantified by ELISA using cytokine-specific DuoSet ELISA systems (R&D Systems) according to the manufacturer's instructions.

## Mice

Female WT C57BL/6j mice were purchased from Janvier Elevage and maintained at the Laboratory Animals Service Center of the University of Zurich under specific pathogen-free conditions. Mice were used at 6–15 weeks of age in age-matched groups. Infected and uninfected animals were kept separately to avoid cross-contamination.

## OPC infection model

Mice were infected sublingually with 2.5 x $10^6$ *C. albicans* yeast cells as described [71], without immunosuppression. In co-infection experiments, mice were infected with $1.25 \times 10^6$ yeast cells of each isolate, resulting in a total of $2.5 \times 10^6$ yeast cells. In 7-day experiments, the body weight of the animals was assessed prior and during the course of infection. For experiments with Dox -inducible *C. albicans* isolates, mice were administered Dox in the drinking water (2 mg/ml) starting from 1 day prior of infection. For determination of the fungal burden, the tongue of euthanized animals was removed, homogenized in sterile 0.05% NP40 in $H_2O$ for 3 minutes at 25 Hz using a Tissue Lyzer (Qiagen) and serial dilutions were plated on YPD agar containing 100 µg/ml Ampicillin. For histology, tissue was fixed in 4% PBS-buffered paraformaldehyde overnight and embedded in paraffin. Sagittal sections (9 µm) were stained with Periodic-acidic Schiff (PAS) reagent and counterstained with Haematoxilin and mounted with Pertex (Biosystem) according to standard protocols. Images were acquired with a digital slide scanner (NanoZoomer 2.0-HT, Hamamatsu) and analysed with NDP.view2.

## Isolation of *C. albicans* from infected tongue tissue

Longitudinally cut tongue halves were placed into 1.5 ml safe-lock tubes containing 800 ml sterile 20% KOH solution and incubated at 60˚C and 300 rpm for at least 30 minutes. Tubes were inverted regularly to prevent sedimentation of tissue fragments. After complete dissolution of the tissue, the suspension was centrifuged for 10 minutes at 6'000 rpm and supernatant was discarded. *C. albicans* cells were suspended in 495 µl Tris-HCl (0.1 M, pH 9) and 5 µl calcofluor white (5-10mg/ml in 0.1 M Tris-HCl, pH 9) to stain the cell wall. After 10 minutes of incubation in the dark, *C. albicans* cells were sedimented for 10 minutes at 6'000 rpm and washed twice with 1 ml Tris-HCl (0.1 M, pH 9). Fungal cell pellets were then resuspended in 20 µl PBS and images with a Leica SP8 inverse confocal microscope.

## Quantification of tongue neutrophils and inflammatory monocytes by flow cytometry

Mice were anesthetized with a sublethal dose of Ketamine (100 mg/kg, Streuli), Xylazin (20 mg/kg, Streuli), and Acepromazin (2.9 mg/kg, Fratro S.p.A.) and perfused by injection of PBS into the right heart ventricle prior to removing the tongue. Tongues were cut into fine pieces and digested with DNase I (200 µg/ml, Roche) and Collagenase IV (4.8 mg/ml, Invitrogen) in PBS at 37˚C for 45–60 min. Single cell suspensions were passed through a 70 µm strainer and stained in ice-cold PBS supplemented with 1% FCS, 5 mM EDTA, and 0.02% NaN3 with LIVE/DEAD Fixable Near-IR Stain (Life Technologies) and fluorochrome-conjugated antibodies against mouse CD45.2 (clone 104), CD11b (clone M1/70), Ly6G (clone 1A8) and Ly6C (clone AL-21). All antibodies were from BioLegend. Stained cells were analysed on a FACS Gallios (Beckman Coulter), or a SP6800 Spectral Analyzer (Sony). The data were analysed with FlowJo software (FlowJo LLC). The gating of the flow cytometric data was performed according to the guidelines for the use of flow cytometry and cell sorting in immunological studies [72]. including pre-gating on viable and single cells for analysis. Absolute cell numbers of lymphocyte populations were calculated based on a defined number of counting beads (BD Bioscience, Calibrite Beads), which were added to the samples before flow cytometric acquisition.

## Preparation of tongue epithelial sheets and isolation of RNA for RNAseq

Epithelial sheets were prepared from infected tongues as described [20]. Briefly, the tongue was cut in half to obtain the dorsal part, which was freed from muscle tissue with a scalpel and

floated on PBS containing 2.86 mg/ml dispase II (Roche) for 60 min with the epithelial side facing upwards to separate the epithelial tissue from the lamina propria. Epithelial sheets were incubated in RNA-later (Sigma) for 1 min immediately after isolation. Three epithelial sheets were pooled per replicate, two replicates were generated for each condition.

For '*in vitro*' samples, *C. albicans* hyphae were induced for 9 hours, harvested using a cell scraper, washed in PBS and rinsed in RNA-later. Two replicates were generated for each condition.

For RNA isolation, epithelial sheets and hyphae pellets were grinded in liquid $N_2$ and processed as described before [73] combining two phase separations and DirectZol RNA Mini-Prep kit (Zymo Research). DNase treatment was performed off-column using a DNA-free Kit (Life Technologies). RNA integrity was determined using a 2100 Bioanalyzer system (Agilent Technologies). RNA was denatured at 70˚C for 2 min prior to analysis. Samples were only included if the RNA integrity value (RIN) was above 7.5 or if two clear peaks for 18S and 25S/28S rRNA were present and no obvious degradation was observed.

## Preparation of cDNA libraries and sequencing

cDNA libraries were generated using SureSelectXT multiplexed sequencing kit with RNA target enrichment for Illumina according to the manufacturer's instructions. In brief, mRNA was purified using poly(A) beads, fragmented, and double-stranded cDNA with ligated adapters was generated. The library was amplified using primers that match the adapters. *C. albicans* specific sequences were enriched using a hybridization technology based on biotinylated baits that cover 6'094 *C. albicans* ORFs but only a negligible fraction of mouse and human cDNA sequences from Ensembl, designed to specifically capture *C. albicans*-derived cDNA [27]. Amplified libraries were incubated with biotinylated baits at 65˚C for 24 h, and hybridized sequences were collected using magnetic streptavidin beads. The collected double-strand cDNA was then amplified and indexed in a separate PCR. For the analysis of RNA-seq samples from *in vitro* induced hyphae, 1% of RNA was used to spike RNA samples from naive mice, which were then subjected to the same enrichment procedure described above. RNA quality, fragment size and cDNA concentration were determined using a fragment analyzer automated CE system (Advanced Analytical) and a Qubit fluorometer (Invitrogen).

For the control on technical issues from the use of biotinylated baits-based technology to capture *C. albicans*-derived cDNA [27] for both isolates SC5314 and 101 as well as the use of SC5314 genome assembly for aligning sequencing data from both isolates, two duplicated RNA-seq samples from SC5314 (963bait and 964bait) and 101 (965bait and 966bait) grown *in vitro* were investigated. Similar percentage of reads were mapped to both isolates with about 98.8% for SC5314 and 98.1% for 101 (**S1 Table**). Similar gene body coverage (distribution of reads on each gene) (**S2A Fig**), count distribution (distribution of gene counts) (**S2B Fig**) and number of expressed genes (RPKM $\geq$1) (**S2C Fig**) were also observed for both isolates. The scatter plots (**S2D Fig**) showed high pairwise correlations among the four control samples, with more than 99.6% of genes in isolate 101 mapped to the SC5314 assembly. Such observations rule out the possible SNPs bias in differential expression analysis between 101 and SC5314.

## RNA-seq data analysis

The sequencing was performed by Lausanne Genomic Technologies Facility (LGTF). cDNA libraries were subjected to cluster generation using the Illumina TruSeq PE cluster kit v3 reagents and sequenced on the Illumina HiSeq 2500 system with TruSeq SBS kit v3 reagents. Sequencing data were processed analogous to a previous study [27]: Data were processed

using Illumina Pipeline software (v1.82). Purity-filtered reads were adapters- and quality-trimmed with Cutadapt (v1.2.1) (Martin, 2011) and filtered for low complexity with Prinseq (v0.20.3) [74]. After alignment against the *Candida albicans* genome SC5314 version A21-s02-m09-r07 using STAR (v2.5) [75,76], htseqcount (v0.5.4p3) was used to summarize the number of read counts per gene locus [77]. Genes with counts fewer than one per million in all samples were removed from the statistical analysis. Data normalization and differential expression analysis were performed in R (v3.2.2), using Bioconductor packages. Normalization of read count data was performed with the R package edgeR using the TMM (trimmed mean of M-values) method [78]. Normalized read count data was transformed to log2 counts per million by voom (implemented in the R package limma) [79]. The hierarchical clustering and principal component analysis were done in R (v3.2.2).

A linear model was built on the transformed data from the following conditions: SC_in_vitro, SC_d1, 101_in_vitro, 101_d1, 101_d3, and 101_d7 (all in duplicates). Differential expression analysis was then performed on the linear model with multiple contrasts representing the differences: (i) between the 101_in_vitro and SC_in_vitro conditions, and (ii) between the 101_d1/3/7 conditions and the SC_d1 condition. P-values produced from the differential analysis were adjusted using the Benjamini & Hochberg correction [80], and adjusted p-values (FDR) < 0.05 were considered significant. Heatmaps were generated with GENE-E (Broad Institute). GO process enrichment analysis was performed with the GO term finder integrated in the Candida Genome Database. The Gene Ontology (GO) term enrichment analysis was performed using DAVID Bioinformatics Resources 6.8 (NIAID, NIH). The enriched GO terms were hierarchically clustered with the semantic similarity between GO terms based on the graph structure of GO (Wang measure) using the R package GOSemSim (v2.12.1) [81] and plotted with the R package ComplexHeatmap (v2.2.0) [82].

## Isolation of fungal RNA from infected keratinocyte monolayers

Monolayers of TR146 cells in 24-well tissue culture plates prepared as described above were infected with $4 \times 10^4$–$10^5$ yeast cells per well. In case of infection with Dox -inducible isolates, 50 μg/ml (in case of SC5314$^{NRG1-OE}$ and SC5314$^{crz2-OE}$) or 10 μg/ml Dox (in case of 101 $^{nrg1\Delta/}$ $_{pTET-NRG1}$, unless stated otherwise) was added. After 24 h of incubation at 37˚C and 5% $CO_2$, RNA was isolated with the RNeasy Mini Kit (Quiagen). Briefly, the cells were lyzed in RLT lysis buffer and homogenized with 0.5 mm glass beads (Sigma) using a Tissue Lyzer (Qiagen) 7 times for 2 minutes at 30 Hz interrupted by 30 sec cooling periods on ice. Cell debris were removed by centrifugation. One volume of 75% ethanol was admixed, and each sample was transferred to a RNeasy spin column. The loaded columns were washed using RW1 buffer and RPE buffer. RNA was eluted in 30 μl RNAse-free water.

## Isolation of host RNA from infected TR146 cells and RHE

Host cell RNA was isolated with TRIZOL according to the manufacturer's instructions.

## RT-qPCR

cDNA was generated by RevertAid reverse transcriptase (Thermo Fisher) and random nonamers. RT-qPCR was performed using SYBR Green (Roche) and a QuantStudio 7 Flex (Life Technology) instrument. All RT-qPCR assays were performed in duplicates and the relative expression levels of each gene was determined by normalization to the housekeeping gene. *hG6PD* was used as housekeeping gene in human and *EFB1* or *ACT1* in *C. albicans*, respectively. Using *EFB1* or *ACT1* as a housekeeping gene provided comparable results. The primers

**Table 3. Primers for detecting host transcripts.**

| Gene | Forward primer (5' -> 3') | Reverse primer (5' -> 3') |
|------|---------------------------|---------------------------|
| *hG6PD* | ATCGACCACTACCTGGGCAA | TTCTGCATCACGTCCCGGA |
| *hIL-6* | TCTGGATTCAATGAGGAGACTTG | CAGGAACTGGATCAGGACTTTTG |
| *hIL-8* | CAAGAGCCAGGAAGAAACCA | GTCCACTCTCAATCACTCTCAG |
| *mActb* | CCCTGAAGTACCCCATTGAAC | CTTTTCACGGTTGGCCTTAG |
| *mIl1b* | CAACCAACAAGTGATATTCTCCAT | GATCCACACTCTCCAGCTGCA |
| *mIl17a* | GCTCCAGAAGGCCCTCAGA | AGCTTTCCCTCCGCATTGA |
| *mS100a9* | GTCCAGGTCCTCCATGATGT | TCAGACAAATGGTGGAAGCA |
| *mIl6* | GAGGATACCACTCCCAACAGACC | AAGTGCATCATCGTTGTTCATACA |
| *mTnf* | CATCTTCTCAAAATTCGAGTGACAA | TGGGAGTAGACAAGGTACAACCC |

**Table 4. Primers for detecting C. albicans transcripts[1].**

| Gene | Forward primer (5' -> 3') | Reverse primer (5' -> 3') |
|------|---------------------------|---------------------------|
| *EFB1* | CATTGATGGTACTACTGCCAC | TTTACCGGCTGGCAAGTCTT |
| *ACT1* | TGCTGAACGTATGCAAAAGG | TGAACAATGGATGGACCAGA |
| *NRG1* | AACCTCAGCCATACCATCAAC | GTAATTAGCCCTGGAGATGGTC |
| *ECE1* | CCAAAATTGCCTGTGCTACTG | CTCTTCATGTTGAATTCTGGAGC |
| *ALS3* | GGTCTCAATCCTATACCACTGC | GGTTGGTGTAATGAGGACGAG |
| *HWP1* | CGGAATCTAGTGCTGTCGTCTCT | CCTTCAAATGTAGAAATAGGAGCAAC |
| *SAP6* | ACTGGGTCTTCTGATTTGTGG | GCAGCTGGAGAATAAGAACCG |
| *PGA25* | TTGTCGGATCTTTTCCCTGG | ATCCTCTTCAGCACTGGAAC |
| *CRZ2* | ACTTCTTCAGCCACGTCATC | CATGCAGTCGAGCAAATCGT |
| *XOG1* | CGACCACATTTCAGTTGCTTG | AGCATTATCGTAAGCACCCTC |
| *PFK1* | TTGAGACACAGAGAATATGGTAGAAG | CTTCTGAAGTGATTGGGTTCAATT |
| *PFK2* | CACCCCTAACGAATTGTACCC | CTGGCTTAGGGTGGGATTTT |
| *FDH1* | GCTAAAGCTCCCAAATTGAAGC | ATGGCAGCGATACCTCTTTC |
| *SAP2* | GGGTTCCTGATGTTAATGTTGATTG | GAAACACCACCAAATCCAACG |

[1] All primers for *C. albicans* transcripts were designed to bind to sequences that are conserved between isolates SC5314 and 101.

used to detect human transcripts are listed in Table 3, those to detect fungal transcripts are listed in Table 4.

## Statistical analysis

Statistical significance was determined by two-tailed unpaired Student's t-test, one- or two-way ANOVA as appropriate using GraphPad Prism 6.0 (GraphPad Software Inc., La Jolla, CA) with *P 0.05; **P 0.01; ***P 0.001; ****P 0.0001.

## Supporting information

**S1 Fig. (related to Fig 3). Isolate 101 persists in the oral mucosa even in presence of SC5314-induced inflammation. A.** The proportion of each isolate 101-mCherry and SC5314-GFP in the infection inoculum 'mix' was analysed prior to infection. **B.** Gating strategy for quantification of neutrophils and inflammatory monocytes in the infected tongue on day 1 post-infection.
(TIF)

**S2 Fig. (related to Fig 4). Read alignments parameters in RNAseq data analysis for SC5314 and 101. A.** Gene body coverage represents the distribution of reads on each gene for each sample of strain SC5314 (963bait and 964bait) and 101 (965bait and 966bait) tested. **B.** Count distribution represents the distribution of gene counts between the four samples. **C.** Number of expressed genes (RPKM ≥1) in each of the categories protein coding, rRNA, mitochondrial and other. **D.** Correlation of gene expression by tested samples of isolates SC5314 and 101. **E.** Numbers of globally up- and downregulated genes between SC5314 and 101 and between the different conditions tested.
(TIF)

**S3 Fig. (related to Fig 4). GO terms associated with differentially regulated C. albicans genes in the infected tongue. A-F.** Enriched GO terms (adjusted p-value < 0.05) on Biological Process and Molecular Function associated with down-regulated (**A, C, E**) and up-regulated genes (**B, D, F**) between isolate 101 on day 1 (**A, B**), day 3 (**C, D**) or day 7 (**E, F**) and SC5314 on day 1 post-infection. GO terms are arranged by hierarchical clustering using Wang-measure semantic similarity. The enrichment adjusted p-value (-log10) is indicated. The horizontal bars represent the number (log2) of differentially regulated genes against the total number of genes associated with each GO term.
(TIF)

**S4 Fig. (related to Fig 4). Confirmation of differentially regulated genes and validation of the altered metabolic signature of isolate 101. A-C.** Isolate 101 and SC5314 were exposed to monolayers of TR146 keratinocytes for 24 hours and expression of the indicated fungal genes was assessed by RT-qPCR. Selected virulence and morphogenesis genes downregulated in isolate 101 are shown in **A**, selected transcription factors and cell wall genes upregulated in isolate 101 are shown in **B**, and differentially regulated genes associated with metabolism are shown in **C**. **D.** Phospholipase activity was assessed after incubation of isolate 101 and SC5314 on Prices' original egg yolk agar for 4 days at 37˚C and 5% $CO_2$. Representative images are shown on the left. The percentage of phospholipase activity (right) was calculated from the ratio of the diameter of the colony (dotted arrow) relative to the diameter of the precipitation zone formed around the colony (solid arrow). **E.** Growth curves for isolates 101 and SC5314 cultured in YPD, YPGlcNAc, YPCasaminoacids or YPBSA medium, respectively. Data are the mean±SD of at least two technical replicates of each sample and representative of at least 3 independent experiments. **F.** Isolates 101 and SC5314 were cultured in $H_2O$ containing Glucose (filled symbols) or GlcNAc (open symbols) and the number of viable fungal cells was quantified at the indicated time points to assess GlcNAc-induced cell death. Each symbol is the mean±SD of duplicate measurements. Data are representative of 3 independent experiments. **G.** Growth curves for the indicated high-damage inducing (solid line) and low-damage inducing (dotted line) *C. albicans* isolates cultured in YPD, YPGlcNAc or YPCasaminoacids medium, respectively. Data are the mean±SD of two technical replicates of each sample and representative of 2 independent experiments.
(TIF)

**S5 Fig. (related to Fig 5): Overexpression of NRG1 in SC5314 restrains its virulence potential *in vitro* and *in vivo*. A.** Overexpression of the *ECE1* gene (from SC5314 under the *TDH3* promoter in isolate 101. *ECE1* expression levels (left panel) and LDH release (right panel) were assessed in four different clones in comparison to the parental isolate 101 after exposure to monolayers of TR146 keratinocyte for 24 hours. **B.-C.** Growth curves of SN76$^{NRG1\text{-}OE}$ (**B**) or SC5314$^{CRZ2\text{-}OE}$ (**C**) and corresponding controls in presence and absence of Dox. Each line is

the mean of 4 samples per condition. **D.** Morphology of SC5314$^{CRZ2\text{-}OE}$ and SC5314$^{CTRL}$ on YPD agar with or without Dox for 2 days. **E.** Monolayers of TR146 keratinocyte were infected with SC5314$^{CRZ2\text{-}OE}$ or SC5314$^{CTRL}$ in presence or absence of Dox and epithelial cell damage was assessed after 24 hours of infection by LDH release assay. Bars are the mean+SEM of 8 samples per condition pooled from 2 independent experiments. **F.-I.** C57BL/6 WT mice were infected sublingually with SC5314$^{CRZ2\text{-}OE}$ or SC5314$^{CTRL}$ and treated or not with Dox. Fungal burden was assessed after 1 day (**F**) or 4 days of infection (**H**). Tongue sections were stained with PAS on day 1 post-infection (**G**). Weight loss and re-gain relative to the pre-infection weight is shown in (**I**). In F and H, each symbol represents one animal, in I each symbol is the mean±SD of 4 animals. (TIF)

**S6 Fig. (related to Fig 6). Reduced expression of NRG1 in isolate 101 increases its pathogenicity. A.** Representative images used for measuring hyphae length shown in Fig 6C for isolates 101$^{nrg1\Delta/\text{NRG1}}$, 101$^{nrg1\Delta/\Delta}$, the parental isolate 101 on monolayers of TR146 keratinocytes in F12 medium. **B-C.** Growth of isolates 101$^{nrg1\Delta/\text{NRG1}}$, 101$^{nrg1\Delta/\Delta}$, the parental isolate 101 and SC5314 on Spider agar (left), YPD agar (middle) and YPGlcNAc agar (right) for 8 days at 30˚C. Colonies were imaged from top (B) or from the side after cutting the agar from top to bottom (C). **D.** Fungal burden in the tongue of C57BL/6 WT mice that were infected sublingually for 1 day with isolate 101$^{nrg1\Delta/\Delta}$ in addition to 101$^{nrg1\Delta/\text{NRG1}}$, the parental isolate 101 and SC5314 as in Fig 6E. (TIF)

**S7 Fig. (related to Fig 7). Suppression of NRG1 expression in isolate 101 via a TET-off strategy drives fungal pathogenicity. A.** Strategy used for generating isolate 101$^{nrg1\Delta/\text{pTET-}NRG1}$ using a Dox shut-off system (pTet-off). Deletion of *NRG1* was performed with a PCR-generated KO cassette. Experimental details are given in the Methods section. FLP, Flippase; RFT, flippase recognition target; NAT1 and SAT1, nourseothricin resistance genes; CatTA, *C. albicans* tetracycline transactivator; HygR, hygromycin resistance. **B.** Functionality of the Tet-off system was tested with a luciferase reporter system. Median values of duplicate measurements for each condition are shown. RLU, relative luminescence units. **C.** Monolayers of TR146 keratinocytes were infected with isolates 101$^{nrg1\Delta/\text{pTET-}NRG1}$ in presence of 0, 1 or 10 μg/ml Dox for 24 hours. Expression of the indicated fungal genes was assessed by RT-qPCR. Bars are the mean+SD of 3 samples per condition from a single experiment. **D.** Representative images used for measuring hyphae length shown in Fig 7D for isolate 101$^{nrg1\Delta/\text{pTET-}NRG1}$ with or without 10 μg/ml Dox on monolayers of TR146 keratinocytes in F12 medium. (TIF)

**S1 Table. (related to Fig 4).** Alignment metrics of two duplicated control RNA-seq samples of SC5314 and 101. (DOCX)

**S2 Table. (related to Fig 4).** Differential gene expression in multiple contrasts between the 101_in_vitro and SC_in_vitro conditions, between SC_day1 and SC_in_vitro conditions, between the 101_d1/3/7 conditions and the 101_in_vitro condition, and between the 101_d1/3/7 conditions and the SC_d1 condition. (XLSX)

# Acknowledgments

The authors would like to thank Noëmi Küng for help with experiments, Françoise Ischer and Danielle Brandalise for technical help with generating *C. albicans* mutants; Selene Mogavero

for assessing Ece1 peptides in ECE1-overexpressing mutants; Ilse Jacobsen for sharing the KOH protocol; the Laboratory Animal Service Center of University of Zürich for animal care; the Laboratory Animal Model Pathology of University for histology; the Lausanne Genomic Technologies Facility of University of Lausanne for RNA sequencing; and members of the LeibundGut-lab for discussions.

## Author Contributions

**Conceptualization:** Dominique Sanglard, Salomé LeibundGut-Landmann.

**Data curation:** Van Du T. Tran.

**Formal analysis:** Christina Lemberg, Kontxi Martinez de San Vicente, Ricardo Fróis-Martins, Simon Altmeier, Salomé LeibundGut-Landmann.

**Funding acquisition:** Christophe d'Enfert, Marco Pagni, Dominique Sanglard, Salomé LeibundGut-Landmann.

**Investigation:** Christina Lemberg, Kontxi Martinez de San Vicente, Ricardo Fróis-Martins, Simon Altmeier, Sarah Mertens, Sara Amorim-Vaz.

**Project administration:** Salomé LeibundGut-Landmann.

**Resources:** Laxmi Shanker Rai, Christophe d'Enfert.

**Supervision:** Dominique Sanglard, Salomé LeibundGut-Landmann.

**Validation:** Salomé LeibundGut-Landmann.

**Visualization:** Van Du T. Tran, Dominique Sanglard, Salomé LeibundGut-Landmann.

**Writing – original draft:** Christina Lemberg, Salomé LeibundGut-Landmann.

**Writing – review & editing:** Christina Lemberg, Kontxi Martinez de San Vicente, Ricardo Fróis-Martins, Simon Altmeier, Van Du T. Tran, Laxmi Shanker Rai, Christophe d'Enfert, Marco Pagni, Dominique Sanglard, Salomé LeibundGut-Landmann.

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
