## [Decision Letter · Decision Letter 0]

3 Nov 2021

Dear LeibundGut-Landman,

Thank you very much for submitting your manuscript "Candida albicans commensalism in the oral mucosa is favoured by limited virulence and metabolic adaptation ." for consideration at PLOS Pathogens. As with all papers reviewed by the journal, your manuscript was reviewed by members of the editorial board and by several independent reviewers. In light of the reviews (below this email), we would like to invite the resubmission of a significantly-revised version that takes into account the reviewers' comments.

As you will read, the reviewers were somewhat split regarding the significance of the observations described in the manuscript. In addition, reviewers 2 and 3 raise important experimental and interpretative concerns that need to be addressed as well.  The concerns of reviewers 2 and 3 are significant and placed the manuscript near the reject point. Therefore, I urge you to carefully consider whether the experimental work required to address these concerns is worth the time that it will take to complete and whether it is feasible.  It is by no means certain that anything less than an extensively revised manuscript will be accepted. 

We cannot make any decision about publication until we have seen the revised manuscript and your response to the reviewers' comments. Your revised manuscript is also likely to be sent to reviewers for further evaluation.

Sincerely,

Damian J Krysan, MD PhD

Associate Editor

PLOS Pathogens

Xiaorong Lin

Section Editor

PLOS Pathogens

Kasturi Haldar

Editor-in-Chief

PLOS Pathogens

orcid.org/0000-0001-5065-158X

Michael Malim

Editor-in-Chief

PLOS Pathogens

orcid.org/0000-0002-7699-2064

Reviewer's Responses to Questions

**Part I - Summary**

Reviewer #1: The study by Lemberg et al. focuses on the oral colonization and pathogenicity determinants of one of the major fungal pathogen of humans, Candida albivans. The work compares side by side the colonization and virulence characteristics of C. albicans isolate strain 101 and the typical reference control SC5314. SC5314, originally a bloodstream isolate that has been lab propagated for many years, is known for its robust virulence phenotypes like avid filamentation and tissue damage. In this study Lemberg et al. once again show that the damaging potential of SC5314 exceeds that of other strains (101) and provide an evidence for it - differences in NRG1, a known hyphal repressor. Alteration of NRG1 levels was sufficient to change the invasiveness in the oral model. Importantly, the study also reveals significant differences in utilization of nutrients by the two strains in support to increasing amount of evidence that C. albicans metabolism and virulence are tightly connected.

Reviewer #2: This is a well-written and interesting manuscript examining the capacity of a clinical strain of C. albicans to colonize murine oral mucosal. Strengths include the genetic modulation on nrg1 in multiple systems and showing a link of the suppressor of filamentation to the colonization phenotype. Multiple other virulence traits are considered. Targeted and nontargeted approaches are complementary. A limitation is the use of a single clinical strain as a comparison to SC5314. Some of the legends should be clarified.

Clarify the data for Figure 2D-H legend. Were experiments performed on multiple days? The way the legend reads, the data included replicate wells from a single experiment.

Figure 4B-C would benefit from additional labeling or explanation. It is not clear what the different colors mean.

Figure 4D: How were the genes selected for display?

Figure 5B-C: Clarify in the legend, should this read right/left (versus top/bottom)?

Figure 6A-B. 7A-B: Were the experiments performed multiple times, or are the data from replicates of a single experiment?

C. albicans killing by neutrophils: This section was in methods, but I didn’t see the experiments in the main text.

Figure S2 legend: Were the strains compared at each time point, or were the various time point for 101 compared to SC5314 on day 1?

Figure S5 legend: For A, what media is used?

Figure S6C: Are the data individual experiments of replicates?

Reviewer #3: In this study by Lemberg and colleagues, the authors seek to gain deeper understanding of the fungal determinants of commensal establishment of Candida albicans in the oral epithelium. Using the previously described oral isolate 101, they characterize its biology and interactions with a mouse model host, in comparison to the pathogenic SC5314 isolate. Compared to SC5314, isolate 101 exhibited shorter filaments in multiple experimental models. Additionally, its expression of virulence-associated transcripts was decreased compared to SC5314, which correlated with less induction of host immunity in the mouse oral epithelium. The authors also found that isolate 101 had increased expression of metabolism-related transcripts. Overall, the manuscript includes a body of work, but a significant portion of the data supports previously understood phenomena, rather than generate novel understanding of the mechanisms supporting 101 persistence.

**Part II – Major Issues: Key Experiments Required for Acceptance**

Reviewer #1: The provided data and the following discussion does not address the following:

1. Is 101 or SC5314 the outlier here? The two strains originate from two different niches (oral vs. blood (with unknown commensal origin)) and it will be important to test another oral isolate shown to cause SC5314-like damage. Will the filamentation and metabolic repertoire of other strain(s) be more 101 or SC5314-like?

2. What is the reason for this difference in NRG1-coordinated effects between the strains? A difference in NRG1 regulation? Can and how this connects to the metabolic adaptation? As NRG1 is relatively well studied it is worthy to take a closer look at the promoter region and provide some more direct evidence for the observed phenotypic differences.

3. Previous reports show that nitrogen utilization by Candida is important in oral infections. The data also suggests that 101 has different efficiency of utilizing nitrogen sources. Testing the repertoire and efficiency of utilization of simple (amino acids/peptides) and more complex (oligopeptides/proteins) nitrogen sources is relatively simple, but would enhance the understanding about fungal persistence in this host niche.

Reviewer #2: None

Reviewer #3: The authors focus on the Candida transcription factor NRG1, whose roles are fairly well understood. Although the authors examine transcripts that are associated with Nrg1 regulation, the function of any of these differentially genes on virulence in the 101 strain is not explored.

There is also a significant experimental/technical problem, in that the 101 transcripts were mapped to the SC5314 genome background. As this strain is quite genetically distant from SC5314, it is likely that the SNPs would alter the ability of a particular read to map to the genome. Recent work has also showed that diverse strains of C. albicans include many novel genes (529L PMID: 31221654 and CHN1 PMID: 34110235), so the transcriptome data needs to be re-analyzed against the appropriate genome reference.

From Figure 3, it is clear that the attenuation of the host immune response to isolate 101 is not responsible for its increased ability to colonize the oral epithelium, as the neutrophil response is actually highest in the mixed culture. Therefore, the focus on the immune response to the Nrg1 mutant strains seems misplaced.

Additionally, the altered metabolism transcriptional signature phenotype is not experimentally followed up, and so the title claim is not well supported by the data.

Therefore, the manuscript is largely descriptive and does not provide significant new knowledge to the field.

**Part III – Minor Issues: Editorial and Data Presentation Modifications**

Reviewer #1: 1. The manuscript is interesting, but at the end remains unclear what it will take for such persistent colonization and low invasiness in the oral cavity to transition into invasive and symptom causing infection. The data shows that this switch is NRG1 coordinated. So, within the oral cavity, when and how mechanistically this switch to the more virulent form will occur? Such discussion (and addressing my major issue #2) will significantly enhance the manuscript.

2. The manuscript needs to be checked throughout for styling issues (spacing, etc.), some grammatically incorrect/overly edited sentences and most importantly - please check if all Materials and Methods are in place as I could not locate some information (like medium composition).

3. I found the introduction relatively difficult to go through as to me the information was not presented in a logical order. Perhaps this is just a personal preference, but I think the flow can be improved. For instance, some of the details in the introduction can be moved to the discussion, which on the other nahd can be expanded. Importantly, a wonderful study by the Jacobsen lab recently published data that metabolic advantage and not filamentation is important for gut pathogenicity. This observation should be discussed.

Reviewer #2: Legends should be clarified as suggested above.

Reviewer #3: Minor Comments:

1. It is unclear how invasion is measured in Figure 2B, and how “percent invasive hyphae” is quantified. The authors should clarify this in the figure legend, and indicate the representative numbers on the images.

2. In the text, for figures such as Figure 2D, the authors only mention that isolate 101 induces less damage. It would be nice to clarify that this is specifically measured by LDH release, and what this specifically means.

3. For Figure 4D, the authors present their transcriptomic analyses as a subset of differentially regulated genes between isolate 101 and SC5314. It would be informative to show global regulation – what percentage of genes are differentially regulated? Additionally, the text describing this figure seems to attempt to fit all of the presented data to the authors’ hypotheses. Are there any differences that are surprising?

4. In Figure 6, the authors show that NRG1 deletion results in increased hyphal length. It would be interesting to see whether this deletion restores hyphal length to that of SC5314.

5. Figure 6I is not referred to in the text.

PLOS authors have the option to publish the peer review history of their article (what does this mean?). If published, this will include your full peer review and any attached files.

Reviewer #1: No

Reviewer #2: No

Reviewer #3: No
---

## [Editor Report · Decision Letter 1]

17 Mar 2022

Dear Salome,

We are pleased to inform you that your manuscript 'Candida albicans commensalism in the oral mucosa is favoured by limited virulence and metabolic adaptation .' has been provisionally accepted for publication in PLOS Pathogens.

Best regards,

Damian J Krysan, MD PhD

Associate Editor

PLOS Pathogens

Xiaorong Lin

Section Editor

PLOS Pathogens

Kasturi Haldar

Editor-in-Chief

PLOS Pathogens

orcid.org/0000-0001-5065-158X

Michael Malim

Editor-in-Chief

PLOS Pathogens

orcid.org/0000-0002-7699-2064

Thank you for your careful, well-reasoned, and experimentally well-supported responses and revisions. I am happy to accept this fine manuscript and congratulate you on an excellent piece of work.
---

## [Editor Report · Acceptance letter]

7 Apr 2022

Dear Dr. LeibundGut-Landmann,

We are delighted to inform you that your manuscript, " *Candida albicans* commensalism in the oral mucosa is favoured by limited virulence and metabolic adaptation. ," has been formally accepted for publication in PLOS Pathogens.

Best regards,

Kasturi Haldar

Editor-in-Chief

PLOS Pathogens

orcid.org/0000-0001-5065-158X

Michael Malim

Editor-in-Chief

PLOS Pathogens

orcid.org/0000-0002-7699-2064